# Demographic reconstruction from ancient DNA supports rapid extinction of the great auk

Jessica E Thomas[1,2‡*], Gary R Carvalho[1†], James Haile[2], Nicolas J Rawlence[3], Michael D Martin[4], Simon YW Ho[5], Arnór Þ Sigfússon[6], Vigfús A Jósefsson[6], Morten Frederiksen[7], Jannie F Linnebjerg[7], Jose A Samaniego Castruita[2], Jonas Niemann[2], Mikkel-Holger S Sinding[2,8], Marcela Sandoval-Velasco[2], André ER Soares[9], Robert Lacy[10], Christina Barilaro[11], Juila Best[12,13], Dirk Brandis[14], Chiara Cavallo[15], Mikelo Elorza[16], Kimball L Garrett[17], Maaike Groot[18], Friederike Johansson[19], Jan T Lifjeld[20], Göran Nilson[19], Dale Serjeanston[21], Paul Sweet[22], Errol Fuller[23], Anne Karin Hufthammer[24], Morten Meldgaard[25], Jon Fjeldså[26], Beth Shapiro[9], Michael Hofreiter[27], John R Stewart[28†], M Thomas P Gilbert[2,4†], Michael Knapp[29†*]

[1]Molecular Ecology and Fisheries Genetics Laboratory, School of Biological Sciences, Bangor University, Bangor, United Kingdom; [2]Natural History Museum of Denmark, University of Copenhagen, Copenhagen, Denmark; [3]Otago Palaeogenetics Laboratory, Department of Zoology, University of Otago, Dunedin, New Zealand; [4]Department of Natural History, University Museum, Norwegian University of Science and Technology, Trondheim, Norway; [5]School of Life and Environmental Sciences, University of Sydney, Sydney, Australia; [6]Verkís Consulting Engineers, Reykjavik, Iceland; [7]Department of Bioscience, Aarhus University, Roskilde, Denmark; [8]Greenland Institute of Natural Resources, Nuuk, Greenland; [9]Department of Ecology and Evolutionary Biology, University of California Santa Cruz, Santa Cruz, United States; [10]Department of Conservation Science, Chicago Zoological Society, Brookfield, United States; [11]Landesmuseum Natur und Mensch Oldenburg, Oldenburg, Germany; [12]Department of Archaeology, Anthropology and Forensic Science, Faculty of Science and Technology, Bournemouth University, Poole, United Kingdom; [13]School of History, Archaeology and Religion, Cardiff University, Cardiff, United Kingdom; [14]Zoological Museum, University of Kiel, Kiel, Germany; [15]Amsterdam Centre for Ancient Studies and Archaeology, University of Amsterdam, Amsterdam, Netherlands; [16]Arqueología Prehistórica, Sociedad de Ciencias Aranzadi, San Sebastián, Spain; [17]Natural History Museum of Los Angeles County, Los Angeles, United States; [18]Institut für Prähistorische Archäologie, Freie Universität Berlin, Berlin, Germany; [19]Gothenburg Museum of Natural History, Gothenburg, Sweden; [20]Natural History Museum, University of Oslo, Oslo, Norway; [21]Humanities Archaeology, University of Southampton, Southampton, United Kingdom; [22]Department of Ornithology, American Museum of Natural History, New York, United States; [23]Independent researcher, Kent, United Kingdom; [24]Department of Natural History, University Museum of Bergen, Bergen, Norway; [25]University of Greenland, Nuuk, Greenland; [26]Center for Macroecology, Evolution and Climate, Natural History Museum of Denmark, University of Copenhagen, Copenhagen, Denmark; [27]Evolutionary Adaptive Genomics, Institute for Biochemistry and Biology, Department of Mathematics and Natural Sciences,

*For correspondence:
drjethomas@hotmail.com (JET);
michael.knapp@otago.ac.nz (MK)

†These authors contributed equally to this work

Present address: ‡Department of Bioscience, College of Science, Swansea University, Swansea, United Kingdom

University of Potsdam, Potsdam, Germany; [28]Faculty of Science and Technology, Bournemouth University, Dorset, United Kingdom; [29]Department of Anatomy, University of Otago, Dunedin, New Zealand

**Abstract** The great auk was once abundant and distributed across the North Atlantic. It is now extinct, having been heavily exploited for its eggs, meat, and feathers. We investigated the impact of human hunting on its demise by integrating genetic data, GPS-based ocean current data, and analyses of population viability. We sequenced complete mitochondrial genomes of 41 individuals from across the species' geographic range and reconstructed population structure and population dynamics throughout the Holocene. Taken together, our data do not provide any evidence that great auks were at risk of extinction prior to the onset of intensive human hunting in the early 16th century. In addition, our population viability analyses reveal that even if the great auk had not been under threat by environmental change, human hunting alone could have been sufficient to cause its extinction. Our results emphasise the vulnerability of even abundant and widespread species to intense and localised exploitation.

## Introduction

The great auk (*Pinguinus impennis*) was a large, flightless diving bird thought to have once numbered in the millions (*Birkhead, 1993*). A member of the family Alcidae in the order Charadriiformes, its closest extant relative is the razorbill (*Alca torda*) (*Moum et al., 2002*). The great auk was distributed around the North Atlantic and breeding colonies could be found along the east coast of North America, especially on the islands off Newfoundland (*Figure 1*). The species also bred on islands off Iceland and Scotland, and was found throughout Scandinavia (Norway, Denmark, and Sweden), with evidence of bone finds existing as far south as Florida and in to the Mediterranean (*Fuller, 1999*; *Grieve, 1885*).

The archaeological and historical records show a long history of humans hunting great auks. In prehistoric times, they were hunted for their meat and eggs by the Beothuk in North America (*Fuller, 1999*; *Gaskell, 2000*), the Inuit of Greenland (*Meldgaard, 1988*), Scandinavians (*Hufthammer, 1982*), Icelanders (*Bengtson, 1984*), in Britain (*Best, 2013*; *Best and Mulville, 2016*), Magdalenian hunter-gatherers in the Bay of Biscay (*Laroulandie et al., 2016*), and possibly even Neanderthals (*Halliday, 1978*). Around 1500 AD intensive hunting began by European seamen visiting the fishing grounds of Newfoundland (*Bengtson, 1984*; *Fuller, 1999*; *Gaskell, 2000*; *Steenstrup, 1855*). Towards the end of the 1700s, the development of commercial hunting for the feather trade intensified exploitation levels (*Fuller, 1999*; *Gaskell, 2000*; *Kirkham and Montevecchi, 1982*). As their rarity increased, great auk specimens and eggs became desirable for private and institutional collections. The last reliably recorded breeding pair were killed in June 1844 on Eldey Island, Iceland, to be added to a museum collection (*Bengtson, 1984*; *Fuller, 1999*; *Gaskell, 2000*; *Grieve, 1885*; *Newton, 1861*; *Steenstrup, 1855*; *Thomas et al., 2017*).

There are scattered records of great auks dating to later than 1844, including in 1848 near Vardø, Norway (*Fuller, 1999*; *Newton, 1861*), and 1852 in Newfoundland (*Fuller, 1999*; *Grieve, 1885*; *Newton, 1861*). BirdLife International/IUCN recognises the last sighting as 1852 (*BirdLife International, 2016a*). However, uncertainty remains about the reliability of all of these later sightings (*Fuller, 1999*; *Grieve, 1885*). There is little doubt that the extensive hunting pressure on the species contributed significantly to its demise. Nevertheless, despite the well documented history of exploitation since the 16th century, it is unclear whether hunting alone could have been responsible for the demise of the great auk, or whether the species was already in decline due to non-anthropogenic environmental changes (*Bengtson, 1984*; *Birkhead, 1993*; *Fuller, 1999*). For example, there is evidence of a decrease in great auk numbers on the eastern side of the North Atlantic, as reflected in a decline in bone finds in England, Scotland, and Scandinavia, which remains unexplained and could have been caused by hunting as well as environmental change (*Bengtson, 1984*; *Best and Mulville, 2014*; *Grieve, 1885*; *Hufthammer, 1982*; *Serjeantson, 2001*). To quote *Bengtson (1984)*, '*In the absence of more detailed information about rate of decline of the*

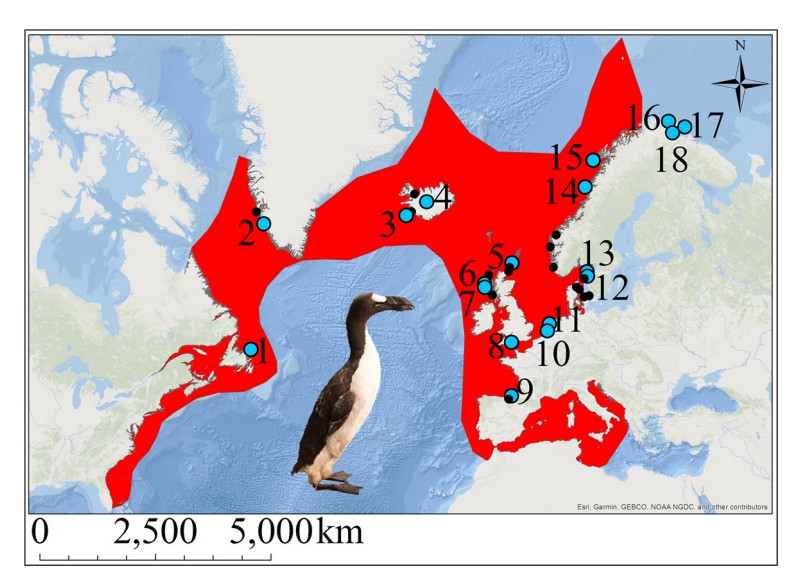

**Figure 1.** The great auk and its former distribution in the North Atlantic. Red shading indicates the geographic distribution of the great auk, as defined by BirdLife International/IUCN (*BirdLife International, 2016a*). Sites marked with blue dots represent samples used in our analyses. Black dots denote other sites from which material was obtained, but for which samples were not sequenced or for which sequences did not pass filtering settings. Numbers associated with blue dots correspond to the following sites: 1: Funk Island (n = 14), 2: Qeqertarsuatsiaat (n = 1), 3: Eldey Island (n = 2), 4: Iceland (n = 5), 5: Tofts Ness (n = 2), 6: Bornais (n = 1), 7: Cladh Hallan (n = 1), 8: Portland (n = 1), 9: Santa Catalina (n = 2), 10: Schipluiden (n = 1), 11: Velsen (n = 1), 12: Sotenkanalen (n = 2), 13: Skalbank Otterön (n = 2), 14: Kirkehlleren (n = 1), 15: Storbåthelleren (n = 1), 16: Iversfjord (n = 1), 17: Vardø (n = 2), and 18: Nyelv (n = 1).

bird populations, hunting pressure and environmental changes, we cannot separate the effects of hunting and that of climate change' (p10).

Reconstructing specific environmental influences on an extinct species can be difficult when there is limited knowledge of the species' biology. However, if the species had been at risk of extinction prior to the onset of intensive hunting in the 16th century, we would expect to see genetic signatures of population decline, including limited genetic diversity and pronounced population structure. In contrast, the lack of an observable loss in genetic diversity during the last few centuries prior to the extinction would be consistent with a rapid demographic decline at the end. At the same time, human hunting alone can only be considered a reasonable explanation for the extinction of the great auk, if population viability analyses show that extinction could have been caused by harvest rates that would have been realistic for the time and circumstances of the harvest.

Here, we examine the drivers of the extinction of the great auk by analysing whole mitochondrial genome (mitogenome) sequences from across its geographic range, population viability, and harvest rates. We combined these with data from GPS-equipped drifting capsules deployed in the North Atlantic, which allow us to suggest potential migration routes among breeding sites.

## Results

### Mitogenome sequence data

Using hybridisation capture combined with high-throughput sequencing, we generated short-read sequence data from 66 bone samples of great auk (See *Supplementary file 1a* for sample information). Following read processing and filtering, 35 samples passed the quality requirements (see Materials and methods) and were suitable for further analysis. In addition to the sequences generated from bones, we included six previously published mitogenome sequences from tissue or feather samples (*Thomas et al., 2017*) (*Supplementary file 1a*).

The combined data set comprised 41 complete mitogenomes, representing individuals from across the former range of the great auk and spanning the period 170–15,000 years before present (ybp). For samples in the final data set, the mean average read length of aligned bases to the reference great auk mitogenome (GenBank accession KU158188.1 [*Anmarkrud and Lifjeld, 2017*]) was 55.12 base pairs (bp), with a range of 41.21–86.95 bp. Unique mitogenome coverage of these samples ranged from 6.39 × to 430.09×, with average coverage of 72.5× (*Supplementary file 1c*). The final alignment length was 16,641 bp, including 9994 bp (after removal of gaps) that were shared across all 41 mitogenomes.

## Genetic diversity and population structure

Haplotype diversity among the great auk mitogenomes was high, with only two individuals yielding identical haplotypes across the 9994 bp covered by all 41 mitogenomes. The two identical sequences differed in age, so that when divided into different age groups, each age group contained a unique set of haplotypes. No reduction of haplotype diversity could be identified in more recent samples (*Figure 2*).

We observed no structure in the distribution of haplotypes using any of our four approaches to reconstruct phylogeographic and temporal relationships among the samples: Bayesian analyses using BEAST (*Appendix 1—figure 1* and *Appendix 1—figure 2*); maximum-likelihood phylogenetic

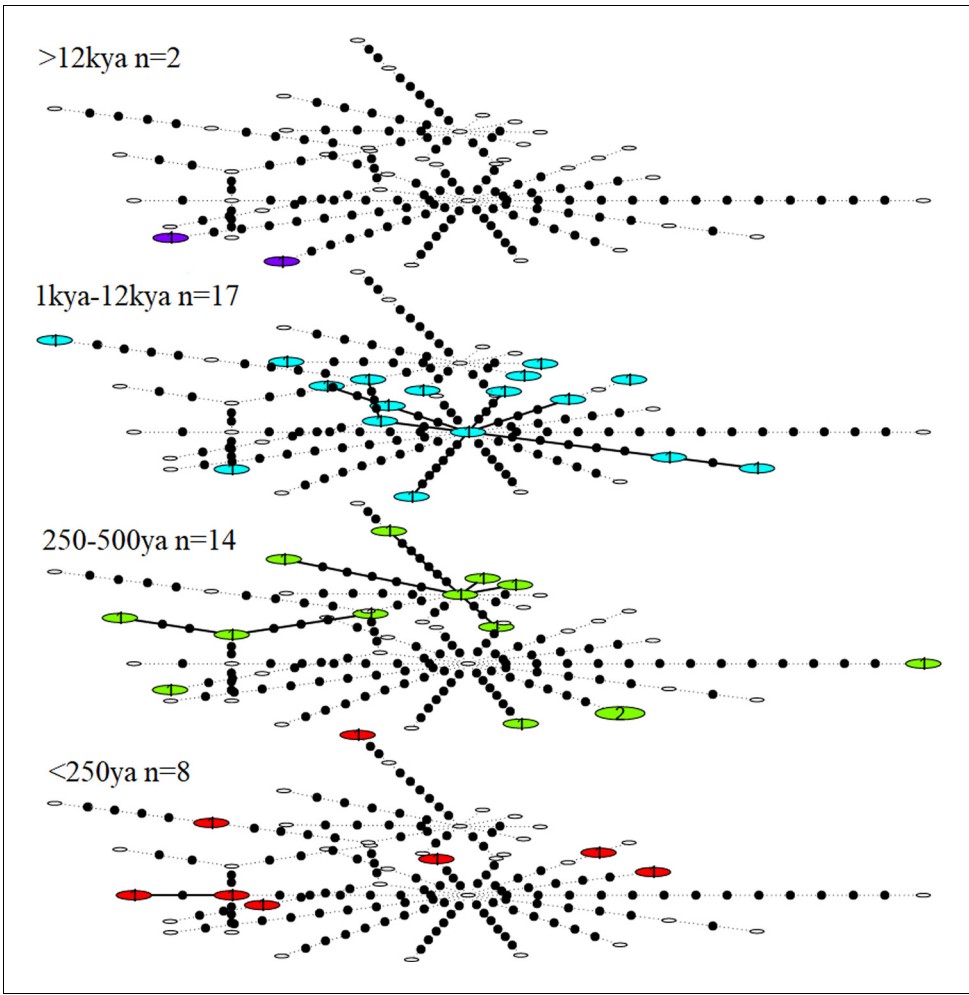

**Figure 2.** Statistical parsimony network showing haplotype diversity of great auk mitogenomes through time. In each age category observed haplotypes are shown in colour, absent haplotypes are shown as empty circles, and mutations between haplotypes are marked as black dots. All samples have been included in this figure.

analysis using RAxML; statistical parsimony network analysis using TempNet (*Figure 2*); and median-joining network analysis using PopART (*Figure 3*).

## Ocean current data

To evaluate potential reasons for the observed lack of population structure, we sourced data from GPS-equipped drifting capsules that had been deployed in the North Atlantic as part of the 'Message in a Bottle' project by Verkís Consulting Engineers. As the great auk was flightless, ocean currents might have influenced its migration patterns. The route taken by the capsules connects some of the main breeding colonies in St Kilda (Scotland), Geirfuglasker/Eldey Island (Iceland), and Funk Island (Canada) (*Figure 4*).

The extrapolation of present-day ocean current data into the past and the interpretation of the data in the context of great auk movements is merely speculative. However, if ocean currents today are comparable to those of past millennia, then the data do at least provide a possible explanation for how great auks travelled across their former range and between breeding colonies (*Figure 4*). A full description of the routes taken by the capsules is provided in Appendix 2.

## Demographic history and effective population size

We reconstructed the demographic history of the great auk using the 25 dated mitogenomes (see Materials and Methods for definition of 'dated' samples) and found support for a constant population size through time, with no evidence of a population decline. Despite having a high haplotype diversity, our samples had a shallow divergence and their most recent common ancestor was dated to 42,188 ybp (95% credibility interval 24,743–84,894 ybp; see Appendix 3). The effective female population size ($N_{ef}$) was estimated at 9558 (95% credibility interval 4548–19,665), assuming a generation interval of 12 years (*BirdLife International, 2016a*). To examine the effect of including the undated samples, we repeated the analysis on the complete data set while accounting for the uncertainty in the ages of the undated samples. This second analysis also yielded support for a constant population size, with an effective female population size of 7331 (95% credibility interval 2477–19,492). Census size ($N_c$) estimates based on the effective population size and the range of known $N_e/N_c$ ratios (*Frankham, 1995*) yielded an expectedly wide range of 12,292–756,346 individuals.

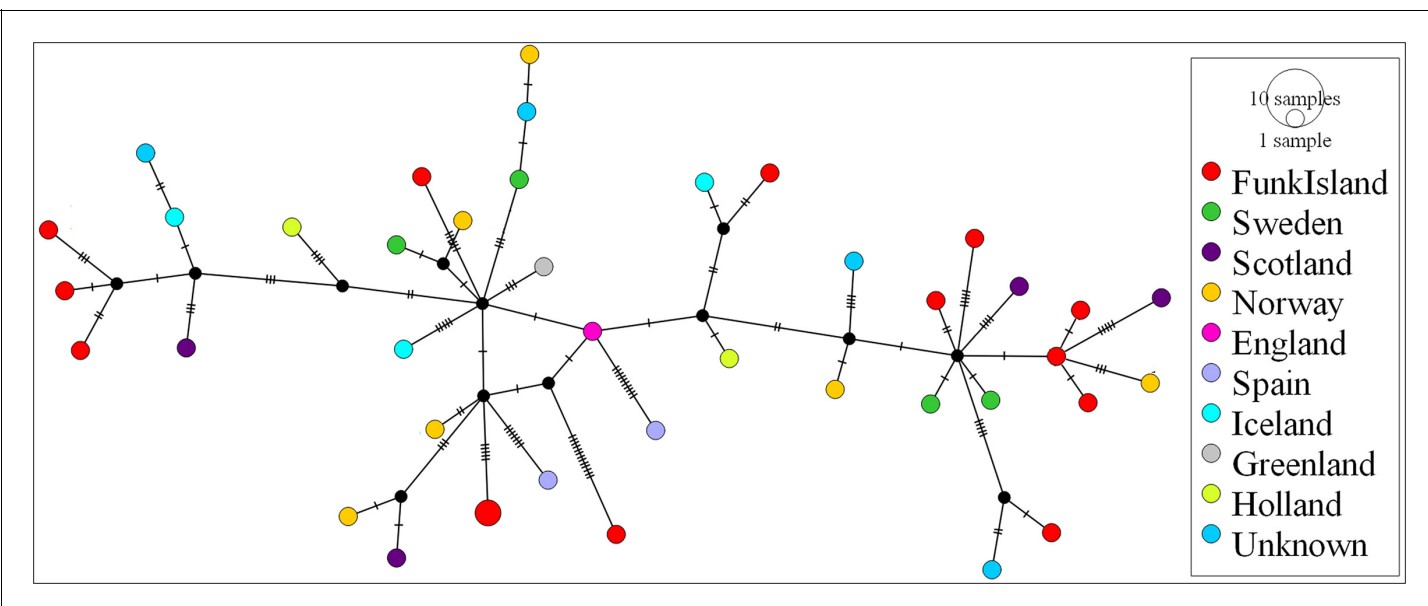

**Figure 3.** Median-joining network of great auk mitogenomes. The network was inferred in PopART18 and shows a lack of phylogeographic structure among the dated and undated samples of great auks. Haplotypes are coloured according to sampling location.

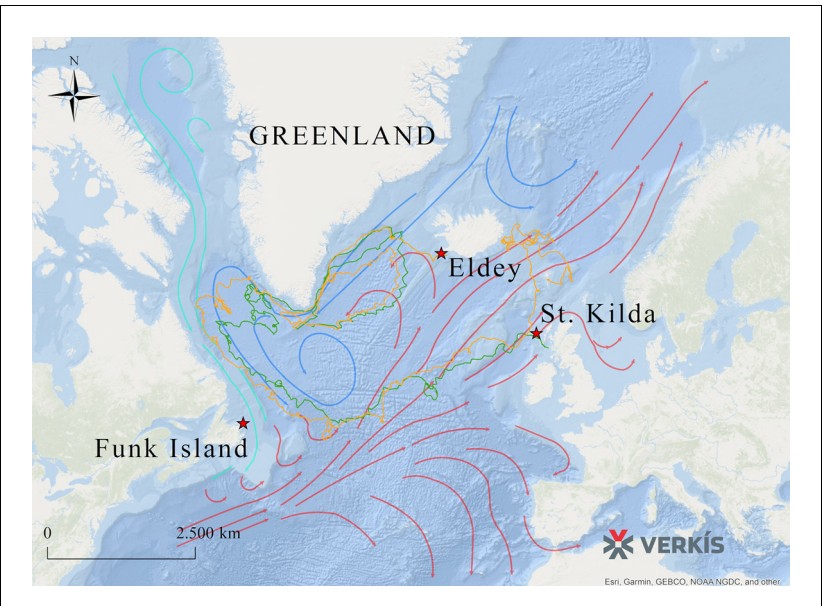

**Figure 4.** Routes taken by GPS capsules in the North Atlantic. The map shows GPS data from two capsules (green and yellow lines). These tracks show possible routes that the great auk might have used to move between colonies, aided by ocean currents, waves, and wind. Legend: Red Star: Known breeding sites of the great auk (Funk Island, New Newfoundland; Eldey Island, Iceland; St Kilda, Scotland). Green line: GPS capsule 1. Yellow line: GPS capsule 2. Pink arrows: Warm sea currents (Gulf Stream and North Atlantic Drift). Dark blue arrows: Cold sea currents (East Greenland Current and Labrador Current).

## Population viability analyses and sustainable harvest rates

To assess the feasibility of a 'hunting-only' scenario of extinction, we used population viability analysis to estimate the proportion of the population that would need to have been harvested in order to cause extinction within 350 years. Population sizes for our simulations were conservatively based on the upper margin of the census size estimates outlined above, consistent with the large census sizes described in historic documents (*Birkhead, 1993*) (see also Appendix 8). The estimate of 756,346 mature birds is slightly below the census size estimates for the great auk's closest relative, the razorbill (*Alca torda*;~1 million mature birds) and significantly below those of common and thick billed murre, also from the Alcidae family (*Uria aalge* and *Uria lomvia*; 3 million mature birds each) (*BirdLife International, 2016a*; *BirdLife International, 2016b*; *BirdLife International, 2016c*; *BirdLife International, 2017*). Given historic reports of millions of great auks (*Birkhead, 1993*) and in order to reduce the risk of underestimating the census size of great auks, we ran simulations for population sizes of 1 million and 3 million mature birds (2 million and 6 million birds total size including juveniles). All simulation settings were 'optimistic' and biased strongly towards survival. This included conservatively high estimates of reproductive success and conservatively low estimates of natural mortality. For a subset of simulations, we also introduced a further, population density dependent, linear reduction of natural mortality to half our already low rates of natural mortality. Furthermore, in order to provide maximum sustainable harvest rate estimates for more 'realistic' settings, we ran simulations using estimates for reproductive success and natural mortality obtained from the razorbill.

We found that under our conservative settings, annual harvest rates up to 9% of the pre-hunting population were sustainable. For example, for a pre-hunting population size of 2 million individuals, this corresponds to an annual harvest rate of 180,000 birds. In contrast, an annual harvest rate of 10% of the pre-hunting population combined with an annual egg harvest rate of 5% led to extinction in a large proportion of our simulations. A harvest rate of 10.5% (egg harvest rate 5%) of the pre-hunting population led to extinction within 350 years in all of our simulations. Assuming a density-dependent reduction of mortality had only a small effect on sustainable harvest rates (*Table 1*). Furthermore, even if no eggs at all were harvested, the population was still at risk of extinction at

**Table 1.** Population viability analysis.

Extinction is defined as 'only one sex remains'. The number of mature individuals was estimated in Vortex 10.2.8.0, assuming a stable age distribution and given our fixed mortality rates. 'Maximum- number of eggs' refers to the number of eggs that would be produced if all mature individuals were breeding. 'Harvest rate' describes the percentage of the population that is harvested annually, with egg harvest rate calculated from the maximum number of eggs in parentheses. 'DD' refers to density-dependent reduction of mortality. 'Number of birds' is the total number of birds killed annually, which was split between the age cohorts (see Appendix 8). 'Number of eggs' is total number of eggs harvested annually.

**Conservative settings**

| Population size (total) | Mature birds (>4 years) | Maximum number of eggs | Harvest rate (% of starting population size) | DD | Number of birds | Number of eggs | Probability of extinction within 350 years |
|---|---|---|---|---|---|---|---|
| 2,000,000 | 1,027,532 | 513,766 | 9 (5) | No | 180,000 | 25,688 | 0.00 |
| 2,000,000 | 1,027,532 | 513,766 | 10 (5) | No | 200,000 | 25,688 | 0.79 |
| 2,000,000 | 1,027,532 | 513,766 | 10 (5) | Yes | 200,000 | 25,688 | 0.22 |
| 2,000,000 | 1,027,532 | 513,766 | 10.5 (5) | Yes | 210,000 | 25,688 | 1.00 |
| 2,000,000 | 1,027,532 | 513,766 | 10.5 (0) | No | 210,000 | 0 | 0.71 |
| 2,000,000 | 1,027,532 | 513,766 | 10.5 (0) | Yes | 210,000 | 0 | 0.19 |
| 6,000,000 | 3,082,594 | 1,541,297 | 9 (5) | No | 540,000 | 77,065 | 0.00 |
| 6,000,000 | 3,082,594 | 1,541,297 | 10 (5) | No | 600,000 | 77,065 | 0.86 |
| 6,000,000 | 3,082,594 | 1,541,297 | 10 (5) | Yes | 600,000 | 77,065 | 0.33 |
| 6,000,000 | 3,082,594 | 1,541,297 | 10.5 (5) | Yes | 630,000 | 77,065 | 1.00 |
| 6,000,000 | 3,082,594 | 1,541,297 | 10.5 (0) | No | 600,000 | 0 | 0.81 |
| 6,000,000 | 3,082,594 | 1,541,297 | 10.5 (0) | Yes | 630,000 | 0 | 0.15 |

**'Realistic' settings**

| Population size (total) | Mature birds (>5 years) | Maximum number of eggs | Harvest rate (% of starting population size) | DD | Number of birds | Number of eggs | Probability of extinction within 350 years |
|---|---|---|---|---|---|---|---|
| 2,000,000 | 1,027,532 | 513,766 | 2 (0) | Yes | 40,000 | 0 | 0.19–0.33 (range across multiple repeat simulations) |

10.5% bird harvest rate, with extinction probabilities between 15% (population size 6 million, density-dependent mortality) and 81% (population size 6 million, no density-dependent mortality, [*Table 1*]). These results were robust to the definition used for extinction. For comparison, when using the much higher mortality rate of the razorbill, with a starting population of 2 million birds and slightly more realistic settings for reproductive age and success, harvest rates are only sustainable up to about 40,000 birds per year even if no eggs are harvested and mortality is gradually reduced to 50% of the starting value as the population density declines (see *Supplementary file 2b*).

## Discussion

Our analyses of the demographic history of great auks support a constant population size within the temporal resolution of our data (back to the most recent common ancestor of all samples 24,000–85,000 ybp). Therefore, we find no evidence of a decline in the population prior to the onset of intensive hunting. We also observed high haplotype diversity across the sampling period, right up to the demise of the species. If the great auk had been at risk of extinction prior to the onset of intensive human hunting, for example as a result of long-term suboptimal habitat or environmental change, we would expect to see genetic evidence of such stress, as for example observed in studies of cave bears (*Stiller et al., 2010*) and bison (*Shapiro et al., 2004*). If, on the other hand, the population declined rapidly, for example as a result of extensive hunting, genetic data would have only very limited power to detect such a decline in a long-lived species. Mitochondrial DNA studies of New Zealand moa found no evidence of a population decline prior to extinction (*Allentoft et al.,*

2014; *Rawlence et al., 2012*) and a study of the endemic Hawaiian Petrel came to a similar conclusion (*Welch et al., 2012*). In fact, even a recent whole-genome study of two extinct New Zealand songbirds (huia and South Island kōkako), which disappeared after human settlement within 700 years, found no genetic evidence of population decline prior to the disappearance of the species (*Dussex et al., 2019*). Therefore, our results are consistent with a rapid decline of great auks. It is important to keep in mind, though, that our results simply indicate that the demise of the great auk was beyond the detection limit of genetic data. They do not necessarily confirm whether the rapid demise that must have taken place prior to extinction started before or after the onset of extensive human hunting, nor do the results provide an indication of whether there was more than one population decline. A localised, unexplained decline in great auk numbers on the eastern side of the North Atlantic over the past 2,000 years, for example, which has been inferred from a decline in bone finds in England, Scotland, and Scandinavia (*Bengtson, 1984*; *Best and Mulville, 2014*; *Grieve, 1885*; *Hufthammer, 1982*; *Serjeantson, 2001*), does not appear to have been severe enough to leave a genetic signature.

The estimated female effective population size is considerably smaller than the census size, which has been estimated to be in the millions (*Birkhead, 1993*). This is noteworthy because it suggests that the species went through a severe bottleneck in the recent past. The shallow divergence of less than 90,000 years between the sequenced individuals suggests a population decline in the late Pleistocene, potentially associated with climate fluctuations. However, the wide 95% credibility intervals of our divergence-time estimates prevent us from narrowing down the cause of the bottleneck to any specific event. In any case, the high percentage of singleton haplotypes in our data, which is characteristic of a population expansion following a bottleneck (*Slatkin and Hudson, 1991*), together with the large census size at the onset of intensive hunting, suggest that the great auk had successfully recovered from the bottleneck.

Our genetic analyses failed to detect any female population structure in space or time, indicating a lack of marked barriers to dispersal among populations across the species' range. This is inconsistent with predictions of limited or no interbreeding between populations from either side of the North Atlantic (*Burness and Montevecchi, 1992*), and suspected regional philopatry in this species (*Bengtson, 1984*; *Montevecchi and Kirk, 1996*). Such a lack of structure is, however, common in seabirds, and has been observed in several relatives of the great auk, such as the thick-billed murre (*Uria lomvia*; no structure within ocean basins) (*Tigano et al., 2015*), common murre (*Uria aalge*; structure in the Atlantic but not in the Pacific) (*Morris-Pocock et al., 2008*), ancient murrelets (*Synthliboramphus antiquus*; no genetic differentiation in the North Pacific) (*Pearce et al., 2009*), and little auk (*Alle alle*; no structure in the Arctic) (*Wojczulanis-Jakubas et al., 2014*). While all of the great auk's closest relatives are capable of flight, which would aid population connectivity, a lack of population structure has similarly been report from some penguin species. For example, little or no population structure has been reported for the emperor penguin (*Aptenodytes forsteri*) (*Cristofari et al., 2016*), chinstrap penguin (*Pygoscelis antarcticus*) (*Mura-Jornet et al., 2018*), and Adélie penguin (*P. adeliae*) (*Gorman et al., 2017*; *Roeder et al., 2001*).

We can only speculate what factors may have driven this lack of population structure, but the data collected from the GPS-enabled drifting capsules are consistent with hypotheses put forward by a number of authors. It has been suggested that migrations occurred in both northward and southward directions between breeding and wintering sites, aided by ocean currents such as the East Greenland Current (*Brown, 1985*; *Meldgaard, 1988*; *Montevecchi and Kirk, 1996*). However, as these preliminary data were only available from two GPS-enabled drifting capsules and as ocean currents may have changed significantly over the past few centuries, the conclusions that we can draw from such data are somewhat limited. Furthermore, it is possible that these currents can change throughout the year. Thus, these data must be considered with caution and pending far more detailed studies of ocean currents in the North Atlantic throughout the year. Nevertheless, high vagility of the great auk is further supported by its ability to track its habitat in response to climate change, as evidenced by archaeological records (*Bengtson, 1984*; *Campmas et al., 2010*; *Meldgaard, 1988*; *Serjeantson, 2001*).

We find no evidence in our genetic data that would suggest that great auk populations were at risk of extinction at the time when human hunting intensified. However, the strength of our conclusions is limited in a number of respects. The mitochondrial genome is only a single genetic marker and our samples were insufficiently preserved to yield nuclear SNP data (Appendix 9), which would

have offered a greater degree of resolution with the potential to detect population structure. Similarly, as a result of limitations in sample preservation and availability, the sample size of 41 is relatively small for population genetic analysis and could have limited our ability to resolve changes in population structure and size.

The key question, therefore, is whether it is at all feasible to assume that the intensive hunting of the 16th–19th centuries alone led to the extinction of the great auk. Our population viability analysis shows that, independent of the population size, harvest rates that would cause extinction under all of the conditions explored in our simulations are well below reasonable estimates of harvest rates as inferred from historical sources. For example, a total population size of 2 million birds corresponds to 1 million mature individuals. This is higher than the upper margin of our census size estimates and is consistent with the census size currently estimated for the great auk's closest relative, the razorbill. At this census size, an annual harvest of 210,000 birds and fewer than 26,000 eggs would have caused the extinction of the great auk within 350 years.

Actual hunting pressure on great auks is likely to have far exceeded 210,000 birds annually. From 1497 AD, when Europeans discovered the rich fishing grounds of Newfoundland, fleets of 300 to 400 ships from various European countries were drawn annually to this region, which is likely to have had the highest population density of great auks (*Bengtson, 1984*; *Steenstrup, 1855*). Fishing stations were set up near colonies of the great auk and other seabirds, and these colonies were heavily exploited (*Pope, 2009*). Great auks were also likely to have been caught by fishing lines and in fishing nets (*Montevecchi and Kirk, 1996*; *Piatt and Nettleship, 1985*; *Piatt and Nettleship, 1987*; *Pope, 2009*). Contemporary reports document a case in which approximately 1000 great auks were caught and killed within half an hour by two fishing vessels off the coast of Funk Island (*Bengtson, 1984*; *Grieve, 1885*). Thus, if each of the 400 vessels in the region spent only half an hour a year harvesting great auks at this rate, that would already correspond to 200,000 birds a year.

At a total population size of 6 million birds, corresponding to the estimated 3 million mature individuals of common murre and thick-billed murre in the North Atlantic, an annual harvest of 630,000 birds and 77,000 eggs would cause certain extinction. Even this number does not appear unrealistically high when considering that great auks were also targeted for the feather trade, with hunters living on Funk Island throughout the summer with the purpose of killing the birds (*Gaskell, 2000*; *Kirkham and Montevecchi, 1982*). Adding to the effects of excessive hunting, the great auk laid only one egg a year, which was not replaced if removed (*Bengtson, 1984*). Thus, replenishing the large number of birds lost annually would have been highly improbable (*Gaskell, 2000*).

Critically, our estimates of harvest rates leading to extinction are likely to be conservatively high, because they are based on some unrealistically optimistic assumptions. For example, our settings assume that 100% of mature birds breed, that they had 100% breeding success, and that their offspring was independent from the time the egg was laid (hence no negative effect of parents being killed). Furthermore, we assumed the lowest natural mortality observed among all alcids for each age class and in some simulations reduced these mortality rates by half when population density declined, thereby considering the positive effects of increased availability of resources and reduced competition. Detrimental effects of small population sizes, such as inbreeding depression, were not included in our simulations. Because very little is known about the biology of the great auk, we chose to use such conservative settings to reduce the risk of underestimating the sustainable harvest rate. However, this brings an increased risk of overestimating the number of birds that could have been sustainably harvested. Using the mortality rate of the razorbill and allowing for more variation in reproductive success (see *Supplementary file 2a*) reduces the sustainable harvest rate for a population of 2 million birds to as few as 40,000 birds per year. However, the razorbill can produce a second egg per season if the first one is lost, so applying razorbill mortality rates to the great auk likely leads to an underestimation of the sustainable harvest rate.

Our conservative simulations require high harvest rates to cause the extinction of the great auk, but these values are largely consistent with harvest rates for present-day species. For example, until recently, between 200,000 and 300,000 murres (*Uria* spp.) were killed legally every year off the eastern Canadian coast (*Wilhelm et al., 2008*). Harvest rates were even higher before the mid-1990s, when between 300,000 and 700,000 thick-billed murres alone were being harvested annually (*Wilhelm et al., 2008*). In Iceland, 150,000 to 233,000 Atlantic puffins were once killed annually, representing about 2–3% of the population. In contrast, 25–30% of the populations of species of black-backed gulls are killed annually (*Merkel and Barry, 2008*). Although current figures for annual

harvest rates of auk species are considerably lower than those given above and continue to decline (e.g.,~25,000 puffins were killed in Iceland in 2016 compared with ~233,000 in 1995 [*Statistics Iceland, 2016*]; also see *Frederiksen et al., 2016*), the harvesting rates required to cause the extinction of the great auk would not be considered excessive even by modern standards.

The roles of humans and environmental changes in causing extinctions have long been debated, not only for the great auk but also for other lost species (*Cooper et al., 2015*; *Lorenzen et al., 2011*; *Shapiro et al., 2004*). In contrast with most studies of Pleistocene extinctions, which have argued for at least some level of climate-driven environmental contributions to species extinction, we have found little evidence that the great auk was at risk of extinction prior to the onset of intensive human hunting. Critically, this does not mean that our study provides unequivocal evidence that humans alone were the cause of great auk extinction. To test this hypothesis, simulations of great auk population dynamics in response to environmental change throughout the Holocene would be required. However, with little information about great auk biology, such simulations would be highly speculative. What our study has demonstrated though, is that human hunting pressure alone was very likely to have been high enough to cause extinction even if the great auk population was not already under threat of extinction through environmental change.

Our findings highlight how industrial-scale commercial exploitation of natural resources have the potential to drive even an abundant, wide-ranging, highly vagile, and genetically diverse species to extinction within a short period of time. This echoes the conclusions drawn for the passenger pigeon (*Murray et al., 2017*), which occurred in enormous numbers prior to its extinction in the early 20th century. Our findings emphasise the need for thorough monitoring of commercially harvested species, particularly in poorly researched environments such as our oceans. This will lay the platform for sustainable ecosystems and ensure the evidence-based conservation management of biodiversity.

## Materials and methods

### Sampling and DNA extraction

We obtained great auk material for ancient DNA (aDNA) analyses from various institutions (*Supplementary file 1a*). Samples were chosen to represent individuals from the major centres of the former geographic distribution of the species (*Figure 1*), spanning as wide a time period as possible (*Supplementary file 1a*). The samples range from about 170 years old to about 13,000–15,000 years old. Sample dates are stratigraphically assigned (archaeological material), based on documented information (e.g., dates on which mounted specimens were killed), or estimated from known site information to give dated constraints (e.g., Funk Island material was collected from the top layers of the islands, so the bones are most likely from individuals killed during the intense hunting period that began ~500 years ago). Bones were sampled via drilling using a Dremel 107 2.4 mm engraving cutter to obtain powdered bone (*Figure 5*) or using a Dremel cutting wheel, which allowed removal of sections of bones that were later powdered using a sonic dismembrator.

All laboratory work prior to polymerase chain reaction (PCR) amplification was carried out in the designated aDNA laboratories of the Natural History Museum of Denmark and the University of Otago. Strict aDNA protocols were followed to avoid contamination. For each DNA extraction and library build, no-template controls were used to test for contamination by exogenous DNA. All post-PCR work was carried out in separate laboratory facilities (*Knapp et al., 2012*).

Genomic DNA was extracted from 20 to 60 mg of bone powder (*Supplementary file 1b*) using the method described by *Dabney et al. (2013)*. In short, the bone powder was digested using an EDTA-based extraction buffer and DNA purified using a Qiagen MinElute column. After washing with ethanol-based wash buffers (Qiagen), the DNA was eluted in TE buffer for storage.

### DNA sequence data

Single-stranded sequencing libraries were prepared from aDNA extracts following the protocol by *Gansauge and Meyer (2013)*, with modifications as described by *Bennett et al. (2014)*. For some samples, double-stranded libraries were also built using the protocol described by *Meyer and Kircher (2010)* (*Supplementary file 1b*). Hybridisation capture was used to enrich libraries for great auk mitochondrial DNA following the MYcroarray MYbaits Sequence Enrichment protocol v2.3.1

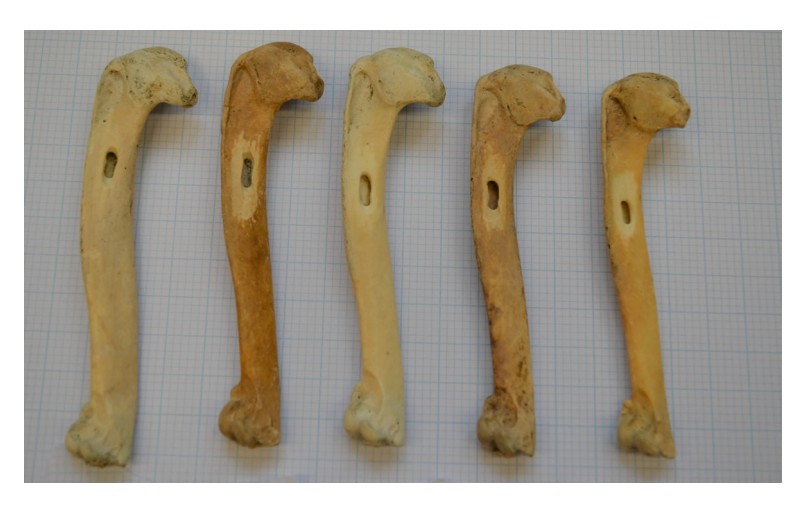

**Figure 5.** Great auk humeri following sampling. Great auk humeri, collected from Funk Island, following sampling to collect bone powder for use in DNA extraction. Bones part of the collection at the American Museum of Natural History (Credit: J. Thomas).

(*MYcroarray MYbaits, 2014*). Bait design details can be found in Appendix 4 and *Appendix 4—figure 1*.

Samples were sequenced on Illumina platforms (HiSeq 2500 and MiSeq; further details in *Supplementary file 1b*) at the Danish National High-Throughput DNA Sequencing Centre or by New Zealand Genomics Limited. Demultiplexing of raw sequence data was performed by the respective sequencing centres. Read processing of demultiplexed sequence data was performed as described by *Thomas et al. (2017)* using the PALEOMIX v1.2.5 pipeline (*Schubert et al., 2014*), details of which can be found in Appendix 5.

## Demographic history analyses

To reconstruct the demographic history of the great auk through time, we performed a Bayesian phylogenetic analysis of the mitogenome sequences from the 25 dated samples ('dated' being defined here as those with associated date information, such as stratigraphically assigned dates; undated refers to those for which there is no associated dating information, such as the Funk Island samples) (*Supplementary file 1e*). The sequence alignment was analysed using BEAST 1.8.4 (*Drummond et al., 2012*). Full details of the BEAST analysis, including details of the data-partitioning scheme, can be found in Appendix 6.

To test hypotheses of constant population size through time vs. population size increase or decline, we compared the marginal likelihoods of constant-size and exponential-growth coalescent tree priors for our data set. The exponential-growth coalescent tree prior with a positive growth rate yielded a higher marginal likelihood than the constant-size tree prior, suggesting that it was the best model of population dynamics in the great auk. However, the posterior distribution of the population growth rate was highly right-skewed with a mode very close to zero, so we conservatively used the constant-size coalescent tree prior for our analysis.

A second analysis was performed in BEAST, in which the 16 undated mitogenomes were included in the data set. A uniform prior of either (0,1000) or (0,5000) was specified for the ages of these mitogenomes, depending on independent information about the context of the samples (*Shapiro et al., 2011*). All other settings and priors matched those used in the analysis of the 25 dated samples. The extended data set was still best described by a constant-size coalescent prior.

## Network analyses

Population structure was investigated by inferring a haplotype network using median joining (*Bandelt et al., 1999*) in PopART (*Leigh and Bryant, 2015*). Genetic diversity through space and time was visualised using statistical parsimony and a temporal haplotype network, as implemented

in TempNet (*Prost and Anderson, 2011*) (see Appendix 7 for details on TempNet age categories and *Supplementary file 1e*).

## Population viability analysis

We performed a population viability analysis using the software Vortex 10.2.8.0 (*Lacy and Pollak, 2014*) in order to estimate the number of great auks that were hunted annually, as well as the rate at which a given intensity of hunting would result in population collapse and extinction. Full details of the simulations performed and parameter justifications can be found in Appendix 8 and *Supplementary file 2a, 2b and 2c*.

## Tracking migration routes using GPS capsules

To achieve a better understanding of the feasibility of great auk movement between colonies of the North Atlantic, we accessed data that were initially generated as part of the 'Message in a Bottle' project by Verkís Consulting Engineers in Iceland. Two GPS-equipped drifting capsules were released on 10th January 2016 from a helicopter around 40 km southeast of the Reykjanes peninsula (southwestern Iceland). Each of the capsules contained a North Star TrackPack GPS tracking device (https://www.northstarst.com/asset-trackers/trackpack/), which uploaded precise location data six times a day for up to two years, through the GlobalStar satellite network.

## Acknowledgements

We are very grateful to the archaeological site directors, sample collectors, curators, and institutions that provided samples for this project. We thank Áki Thoroddsen (Verkís) for producing *Figure 4*. Funding was provided through a NERC PhD Studentship (NE/L501694/1) to JET, ERC Consolidator Award (681396-Extinction Genomics) to MTPG, the Genetics Society-Heredity Fieldwork Grant, and European Society for Evolutionary Biology–Godfrey Hewitt Mobility Award to JET. MK is supported by a Rutherford Discovery Fellowship from the Royal Society of New Zealand. We thank members of the Molecular Ecology and Fisheries Genetics Laboratory at Bangor University, EvoGenomics and GeoGenetics at University of Copenhagen, and the Biological Anthropology group at the University of Otago for guidance in the laboratory and on data analysis and for useful discussions. Sequencing was provided by The Danish National High-Throughput DNA Sequencing Centre and New Zealand Genomics Limited.

## Additional information

### Competing interests

Arnór Þ Sigfússon: Arnór Þ Sigfússon is affiliated with Verkís Consulting Engineers. The author has no financial interests to declare. Vigfús A Jósefsson: Vigfús A Jósefsson is affiliated with Verkís Consulting Engineers. The author has no financial interests to declare. The other authors declare that no competing interests exist.

### Funding

| Funder | Grant reference number | Author |
| --- | --- | --- |
| NERC Environmental Bioinformatics Centre | NE/L501694/1 | Jessica E Thomas |
| European Research Council | 681396-Extinction Genomics | M Thomas P Gilbert |
| Genetics Society | Heredity Fieldwork Grant | Jessica E Thomas |
| European Society for Evolutionary Biology | Godfrey Hewitt Mobility Award | Jessica E Thomas |
| Royal Society of New Zealand | Rutherford Discovery Fellowship | Michael Knapp |

The funders had no role in study design, data collection and interpretation, or the decision to submit the work for publication.

### Author contributions
Jessica E Thomas, Conceptualization, Data curation, Formal analysis, Funding acquisition, Investigation, Visualization, Writing—original draft, Project administration, Writing—review and editing; Gary R Carvalho, Conceptualization, Supervision, Writing—original draft, Project administration, Writing—review and editing; James Haile, Mikkel-Holger S Sinding, Marcela Sandoval-Velasco, Investigation, Writing—original draft; Nicolas J Rawlence, Conceptualization, Writing—original draft, Writing—review and editing; Michael D Martin, Formal analysis, Investigation, Writing—original draft; Simon YW Ho, Data curation, Software, Formal analysis, Investigation, Visualization, Writing—original draft, Writing—review and editing; Arnór Þ Sigfússon, Vigfús A Jósefsson, Investigation, Visualization, Writing—original draft; Morten Frederiksen, Jannie F Linnebjerg, Christina Barilaro, Juila Best, Dirk Brandis, Chiara Cavallo, Mikelo Elorza, Kimball L Garrett, Maaike Groot, Friederike Johansson, Jan T Lifjeld, Göran Nilson, Dale Serjeanston, Paul Sweet, Errol Fuller, Anne Karin Hufthammer, Morten Meldgaard, Jon Fjeldså, Resources, Writing—original draft; Jose A Samaniego Castruita, Jonas Niemann, Data curation, Software, Writing—original draft; André ER Soares, Formal analysis, Writing—original draft; Robert Lacy, Software, Formal analysis, Investigation, Writing—original draft, Writing—review and editing; Beth Shapiro, Formal analysis, Writing—original draft, Writing—review and editing; Michael Hofreiter, John R Stewart, Conceptualization, Supervision, Writing—original draft, Writing—review and editing; M Thomas P Gilbert, Conceptualization, Supervision, Funding acquisition, Investigation, Writing—original draft, Project administration, Writing—review and editing; Michael Knapp, Conceptualization, Formal analysis, Supervision, Funding acquisition, Investigation, Visualization, Writing—original draft, Project administration, Writing—review and editing

### Author ORCIDs
Jessica E Thomas  https://orcid.org/0000-0002-9043-646X
James Haile  http://orcid.org/0000-0002-8521-8337
Simon YW Ho  https://orcid.org/0000-0002-0361-2307
Michael Knapp  https://orcid.org/0000-0002-0937-5664

### Decision letter and Author response
Decision letter https://doi.org/10.7554/eLife.47509.SA1
Author response https://doi.org/10.7554/eLife.47509.SA2

## Additional files

### Supplementary files
• Source data 1. Nuclear SNP bait design.

• Supplementary file 1. Sample Information. *Supplementary file 1a* Sample information for all samples collected. Information listed shows institution name and number where sample was sourced, the site location and country where sample was discovered (if known), and any associated date/age information, if known. Those highlighted indicate samples that ultimately passed the filtering settings and were used in the final analysis. Asterisks indicate samples from *Thomas et al. (2017)*. *Supplementary file 1b* Lab process table for all samples collected. Table includes information on sample type, weight used in extraction, which library build method was used, if hybridization capture was used, and which type of sequencing was performed. Those highlighted indicate samples that ultimately passed the filtering settings and were used in the final analysis. Asterisks indicate samples from *Thomas et al. (2017)*. *Supplementary file 1c* PALEOMIX summary data for mitogenome samples. Summary statistics table from all great auk samples sent for sequencing. Library type: PE = Paired end, SE = Single end, *=both. Samples highlighted were used in final analysis. *Supplementary file 1d* GenBank accession numbers. GenBank accession numbers for samples used in analysis. *Supplementary file 1e* Age information for samples used in analysis. Age information for samples used in the BEAST and TempNet analyses.

• Supplementary file 2. Population Viability Analysis Settings.*Supplementary file 2a* Settings used in Population Viability Analysis. Details of the settings used for Population Viability Analysis performed in Vortex 10.2.8.0. Information displayed corresponds to the various setting sections in the software and the variables that were changed. Further details on justification for these settings can be found in Appendix 8. *Supplementary file 2b* Details of mortality rates used in Population Viability Analysis. Details of the mortality rate settings used in Population Viability Analysis performed in Vortex 10.2.8.0, showing formula information for including density-dependent change and additional justification. *Supplementary file 2c* Harvest rate calculations. Example of how harvest rates of birds and eggs were calculated for the Population Viability Analysis.

• Transparent reporting form

## Data availability

Sequence data are available on NCBI GenBank under the Popset IDs 1735592912 and 1208276182.

The following dataset was generated:

| Author(s) | Year | Dataset title | Dataset URL | Database and Identifier |
|---|---|---|---|---|
| Thomas JE, Carvalho GR, Haile J, Rawlence NJ, Martin MD, Ho SYW, Sigfusson AÞ, Josefsson VA, Frederiksen M, Linnebjerg JF, Samaniego Castruita JA, Niemann J, Sinding M-HS, Sandoval-Velasco M, Soares AER, Lacy R, Barilaro C, Best J, Brandis D, Cavallo C, Elorza M, Garrett KL, Groot M, Johansson F, Lifjeld JT, Nilson G, Serjeanston D, Sweet P, Fuller E, Hufthammer AK, Meldgaard M, Fjeldsa J, Shapiro B, Hofreiter M, Stewart JR, Gilbert MTP, Knapp M | 2019 | Pinguinus impennis mitochondrion, partial genome | https://www.ncbi.nlm.nih.gov/popset/?term=1735592912 | NCBI Popset, 1735592912 |

The following previously published dataset was used:

| Author(s) | Year | Dataset title | Dataset URL | Database and Identifier |
|---|---|---|---|---|
| Thomas JE, Carvalho GR, Haile J, Martin MD, Castruita JAS, Niemann J, Sinding MS, Sandoval-Velasco M, Rawlence NJ, Fuller E, Fjeldsa J, Hofreiter M, Stewart JR, Gilbert MTP, Knapp M | 2017 | Pinguinus impennis mitochondrion, complete genome | https://www.ncbi.nlm.nih.gov/popset/?term=1208276182 | NCBI PopSet, 1208276182 |

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

Appendix 1

## Phylogenetic trees

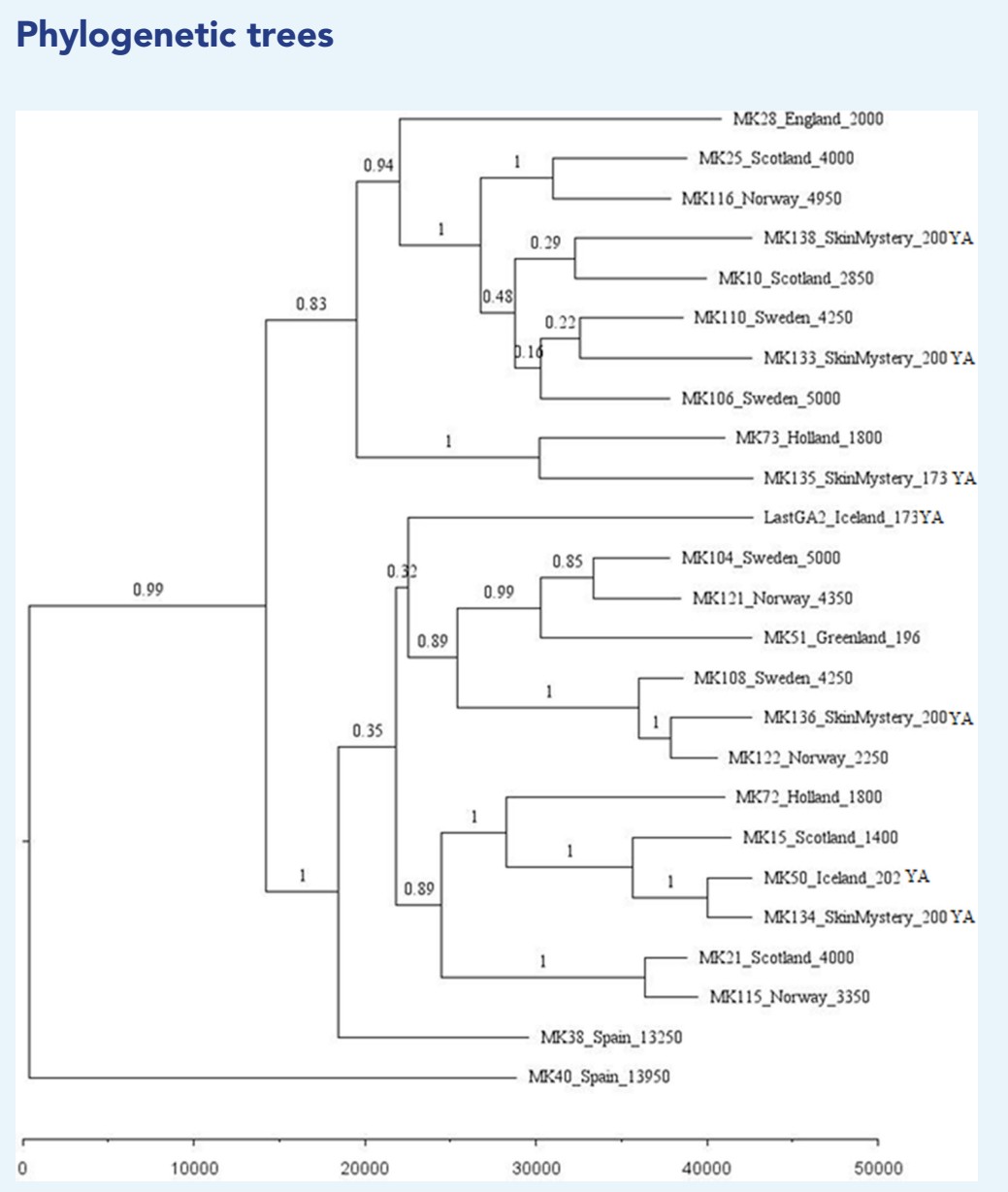

**Appendix 1—figure 1.** Phylogenetic tree showing the relationships among dated mitogenomes from the great auk. This maximum-clade-credibility tree was inferred by Bayesian analysis in BEAST. Nodes are labelled with posterior probabilities. The tree is drawn to a timescale, as indicated by the horizontal scale bar. Samples included in the analysis are those with associated date information (see *Supplementary file 1e*). For samples with a stratigraphically assigned date the median age has been used. Tip labels give the sample names, sampling locations, and sample ages (years before present, with the exception of mounted specimens labelled YA- years ago).

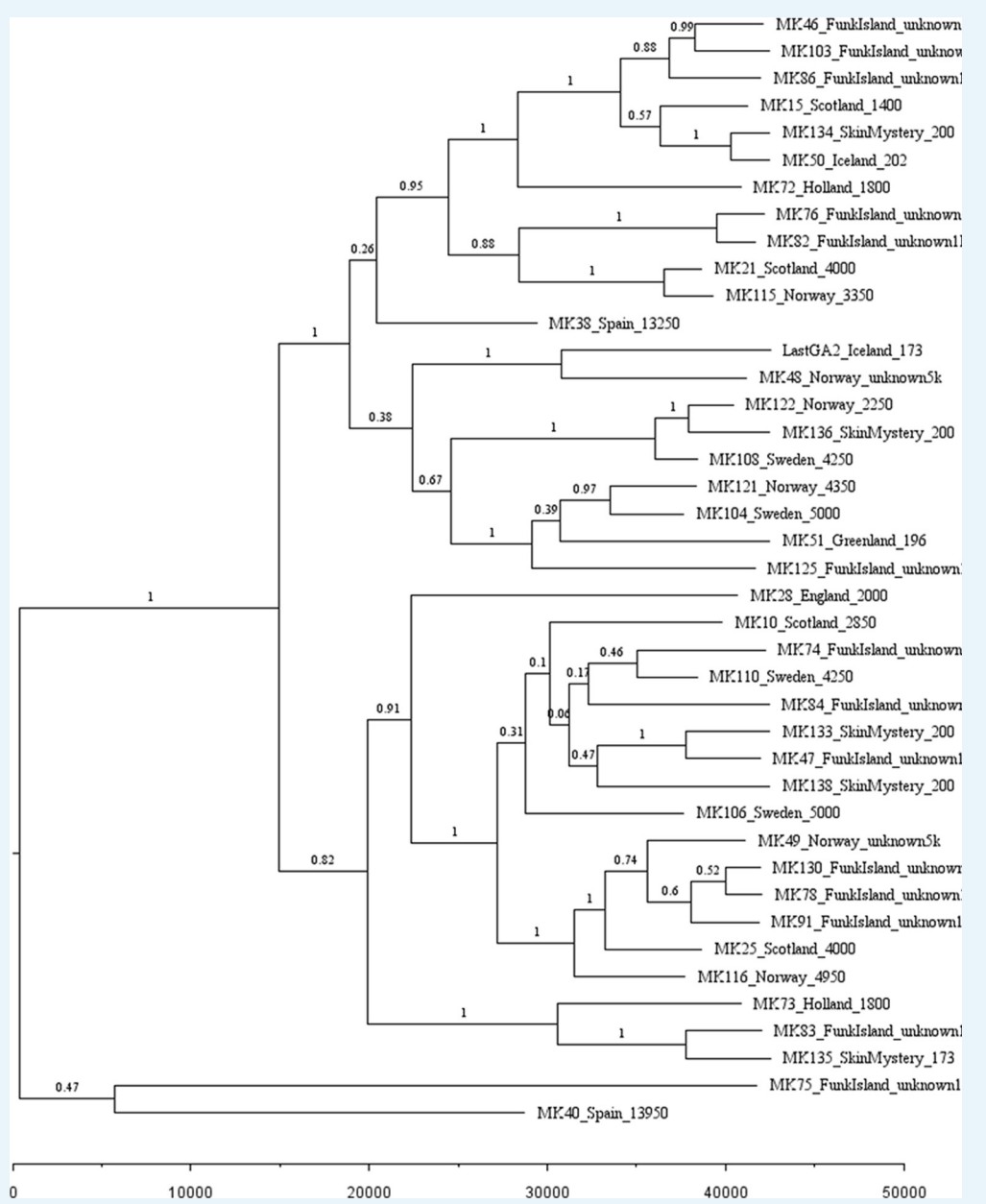

**Appendix 1—figure 2.** Phylogenetic tree showing the relationships among dated and undated mitogenomes from the great auk. This maximum-clade-credibility tree was inferred by Bayesian analysis in BEAST. Nodes are labelled with posterior probabilities. The tree is drawn to a timescale, as indicated by the horizontal scale bar. Samples included in the analysis are those with and without associated date information (*Supplementary file 1e*). For samples with a stratigraphically assigned date the median age has been used. Tip labels give the sample names, sampling locations, and sample ages (years before present, with the exception of mounted specimens labelled YA- years ago).

**Appendix 2**

## GPS-equipped drifting capsules: Full result

Following release, easterly winds prevailed and the two GPS-equipped drifting capsules drifted westwards past the tip of the Reykjanes peninsula and past Eldey Island (*Figure 4*). Over the next two weeks, the capsules drifted towards Greenland and when located near the continental shelf started drifting southwards along the coast. The capsules then followed the track of the Icelandic Low, a low-pressure area found between Iceland and Southern Greenland in winter.

The Icelandic Low took the capsules in an anti-clockwise circle back towards Iceland, and onward again towards the west coast of Greenland. The Icelandic Low weakens in summer, so in late April the capsules turned westwards past the southern tip of Greenland and into the Labrador Sea. In summer, they drifted slowly towards the Labrador coast until the beginning of August when they started drifting south-eastwards along the coast of Labrador and Newfoundland and past Funk Island and around 500 km east. By the end of October, the capsules start to follow the trail of the winter low pressures across the Atlantic.

At the beginning of January, capsule one drifted eastwards, around 50 km south of St Kilda and came ashore on the island of Tiree (15.01.2017). Capsule two drifted northwards, passing around 70 km west of St Kilda and west of the Faeroes towards Iceland. In early March, the capsule was around 20 km from the east coast of Iceland when it turned eastwards and then towards south by the beginning of April. It drifted towards the Faeroes where it came ashore on the island of Sandoy (13.05.2017).

The forces driving the capsules are currents, wind, and waves. The capsules got trapped in the Iceland Low where the wind direction is in a counter clockwise circle in winter in the Denmark Strait. In spring, when the Iceland Low starts dissolving, they pass Cape Farewell and, in summer, they drift slowly in calmer summer winds and followed the cold current towards the Labrador coast and then along the coast of Labrador and Newfoundland. In autumn, they hit the path of lows crossing the Atlantic as well as following the warmer Gulfstream. In spring, when at the east coast of Iceland, the weather was calmer and thus capsule two drifted slowly towards and then away from the coast and ended up in the Faeroes.

## Appendix 3

### Molecular dating

Estimating the age of the most recent common ancestor (MRCA) of all our samples is not essential to understanding the causes of the extinction of the great auk, but can help with the interpretation of our reconstructions of population dynamics. Our data set is unable to yield reliable information about population dynamics beyond the MRCA of all samples. To infer the evolutionary rate and timescale, we performed a Bayesian phylogenetic analysis of the mitogenome sequences from the 25 dated samples. The analyses were conducted using the same settings and data-partitioning scheme as described in the Methods for our Bayesian phylogenetic analyses.

The sequence alignment was analysed using BEAST 1.8.4 (*Drummond et al., 2012*). The evolutionary timescale was estimated using a strict clock model, with the sampling times of the mitogenomes serving as calibrations for the clock (*Rambaut, 2000*). Furthermore, to test for the presence of temporal structure in the data set, we performed a date-randomisation test (*Ramsden et al., 2008*). We estimated mutation rates from 20 replicate data sets in which the sampling times were permuted and compared these with the rate estimate from the original data set. Two different criteria can be used to determine whether the data set has sufficient temporal structure for generating a reliable estimate of the mutation rate (*Duchêne et al., 2015*): if the mean or median estimate from the original data set is not contained within the 95% credibility intervals of the rate estimates from the date-randomised replicates (CR1), or if the 95% credibility intervals of the rate estimates from the date-randomised replicates do not overlap with the 95% credibility interval of the rate estimate from the original data set (CR2).

For comparison, we used two additional methods to estimate the mutation rate. First, we used TempEst (*Rambaut et al., 2016*) to estimate the mutation rate using regression of root-to-tip distances against sampling times. Second, we analysed the data using least-squares dating in LSD (*To et al., 2016*). For both of these methods, a phylogram was estimated from the dated mitogenome sequences using maximum likelihood in RAxML 8 (*Stamatakis, 2014*). Rooting of the tree was inferred by maximising the R-squared value in TempEst and by minimising the objective function in LSD.

Our Bayesian phylogenetic analysis of the dated mitogenomes produced a posterior median estimate of 42,188 years (95% credibility interval 24,743–84,894 years) for the age of the most recent common ancestor. The median posterior estimate of the mutation rate was $2.74 \times 10^{-8}$ mutations/site/year (95% credibility interval $9.83 \times 10^{-9}$–$4.53 \times 10^{-8}$). The data set showed some evidence of temporal structure, passing the more lenient criterion CR1 but not the more stringent CR2 of the date-randomisation test (*Appendix 3—figure 1*; *Duchêne et al., 2015*). Thus, with all caution required given the limited temporal structure in our data, our inference of a constant population size for the great auks should be reliable reaching back to the late Pleistocene. However, our data set is not likely to be suitable for drawing strong conclusions about population dynamics of the great auk beyond the last glacial period.

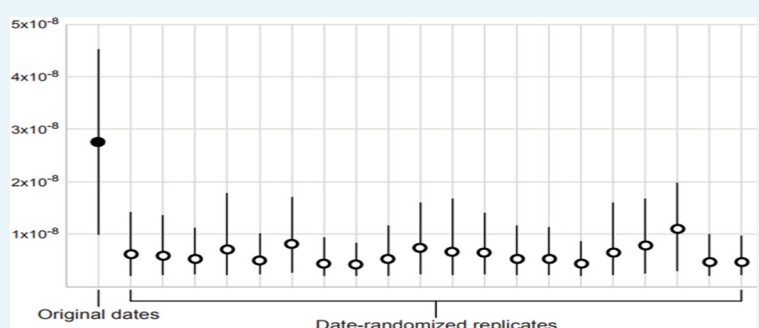

**Appendix 3—figure 1.** Date-randomisation test for temporal structure in dated mitogenome sequences. The filled circle indicates the median posterior estimate of the mutation rate from the original data set, whereas the empty circles show the median posterior estimates from 20 date-randomised replicate data sets. The 95% credibility intervals (vertical black lines) of the estimates from the date-randomised replicates do not overlap with the median estimate from the original data set, providing some evidence of temporal structure in the data set (criterion CR1). However, the 95% credibility intervals of the estimates from the date-randomised replicates overlap with the 95% credibility interval of the estimate from the original data set, indicating that the data set does not meet the more stringent criterion CR2.

## Appendix 4

### Bait design

100mer mitochondrial DNA baits (MYcroarray MYbaits) with 50 bp tiling were designed using a hybrid reference mitogenome. This was constructed using the mitogenome from killdeer (*Charadrius vociferus*; assembled from whole genome data, BioProject: PRJNA212867 [*Zhang et al., 2014*]), with orthologous gene regions replaced by those of great auk where available (GenBank: AJ242685), and those from the razorbill (*Alca torda*; GenBank accessions AJ301680, EF380281, EF380318, and X73916) when great auk data were unavailable (*Appendix 4—figure 1*).

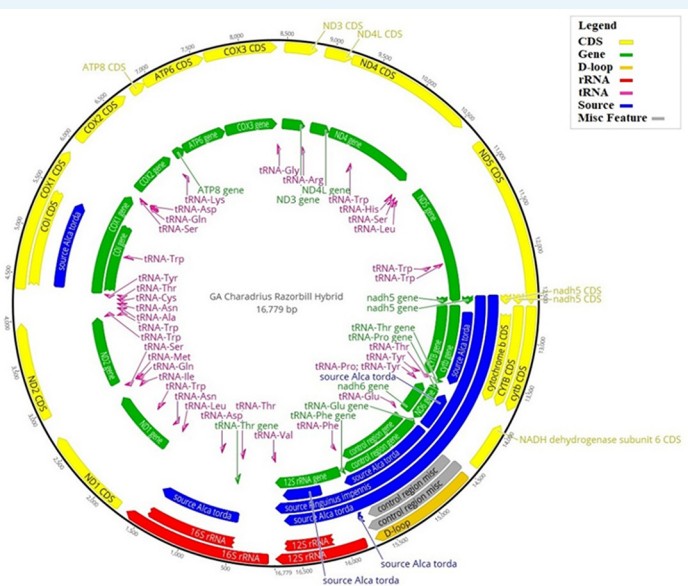

**Appendix 4—figure 1.** Hybrid reference mitogenome used for bait design. Illustration of the hybrid reference mitogenome constructed using the killdeer (*Charadrius vociferous*) mitogenome, with orthologous gene regions replaced by those of the great auk (*Pinguinus impennis*), or razorbill (*Alca torda*), when great auk data were unavailable. Annotations correspond to the various regions of the mitogenome: those in blue show where great auk or razorbill genes have been used; yellow corresponds to coding regions; green shows all gene regions; the D-loop is shown in gold; rRNA regions are in red; tRNA regions are in pink; and any miscellaneous features are in grey. The numbers on the outer black circle correspond to the base position of the mitogenome.

## Appendix 5

### Read processing

Read processing was performed using the PALEOMIX v1.2.5 pipeline (*Schubert et al., 2014*). The procedure included software tools to remove adapters, filter bases based on quality (AdapterRemoval v2.1.7 [*Lindgreen, 2012*; *Schubert et al., 2016*]), and map reads to the reference mitogenome (Burrows-Wheeler Aligner v0.5.10 [*Li and Durbin, 2009*]). At the time of these analyses, a great auk mitogenome had been published (GenBank: KU158188.1 [*Anmarkrud and Lifjeld, 2017*]), and was thus available for the mapping assembly of our mitogenomes rather than mapping against the composite mitogenome used for bait design (see above).

PCR duplicates were removed using MarkDuplicates within Picard v1.8.2 (*Broad Institute, 2019*) and the rmdup function within SAMtools (*Li et al., 2009*). The Genome Analysis Toolkit (GATK) v3.6.0 was used to correct for misaligned reads to the reference mitogenome using the RealignerTargetCreator and IndelRealigner functions (*McKenna et al., 2010*). Finally, MapDamage2 (*Jónsson et al., 2013*) was employed to rescale base-quality scores according to their probability of being damaged, thereby removing residual aDNA damage patterns. The UnifiedGenotyper algorithm within GATK v3.6.0 was used to determine haploid genotypes for individual samples.

Consensus sequences were produced using the following filtering settings. The per-individual read depth was set to include only bases with a minimum of 3-fold coverage. Bases called for the consensus sequence had to be present at a frequency higher than 33%. To be included in the final alignment, no more than 33% of bases could be missing from the consensus sequence of an individual compared with the reference sequence. All bases failing to meet these criteria were called as 'N' (*Chang et al., 2017*).

Following read processing, the sequences were aligned using Seaview v4.0 (*Gouy et al., 2010*) with the algorithm *Muscle -maxiters2 -diags.* The alignment was manually checked for errors using BioEdit v7.2.5 (*Hall, 1999*). Tablet v1.16.09.06 (*Milne et al., 2013*) was used to view the rescaled Binary Alignment Map (BAM) file for each sample.

Sequence data from all samples included in the analysis have been deposited in GenBank. The GenBank accession numbers for samples included in the final analysis can be found in *Supplementary file 1d*.

## Appendix 6

## Population dynamics analysis: Settings, partitioning schemes and further details

Six partitioning schemes were compared for the data, varying in the degree of partitioning and the resulting number of data subsets (*Appendix 6—table 1.*). For each data subset, the best-fitting model of nucleotide substitution was selected using the Bayesian information criterion in Modelgenerator (*Keane et al., 2006*). A partitioning scheme with six data subsets provided the best fit to the data.

**Appendix 6—table 1.** Marginal likelihoods of six partitioning schemes and two tree priors for the 25 dated mitogenomes.

| Partitioning scheme[a] | Marginal likelihood[b] | |
|---|---|---|
| | Constant size | Exponential growth |
| Unpartitioned | −24,151.6 | −24,143.6 |
| two subsets: (CR rRNA tRNA) (PC1 PC2 PC3) | −24,222.3 | −24,212.4 |
| three subsets: (CR) (rRNA tRNA) (PC1 PC2 PC3) | −24,162.4 | −24,150.1 |
| four subsets: (CR) (rRNA tRNA) (PC1 PC2) (PC3) | −23,659.7 | −23,647.5 |
| five subsets: (CR) (rRNA tRNA) (PC1) (PC2) (PC3) | −23,248.7 | −23,235.9 |
| six subsets: (CR) (rRNA) (tRNA) (PC1) (PC2) (PC3) | −23,229.1 | −23,217.5 |

[a]Components of the mitogenome are the ribosomal RNA genes (rRNA), transfer RNA genes (tRNA), three codon positions of the protein-coding genes (PC1, PC2, and PC3), and the control region (CR). [b]Marginal likelihoods were estimated by stepping-stone sampling with 25 path steps, each with a chain length of 2,000,000 steps.

Constant-size and exponential-growth coalescent tree priors were compared for the data. Analyses using a skyride coalescent prior (*Minin et al., 2008*) were attempted but invariably failed to converge, which strongly suggested overparameterisation. The marginal likelihood was computed for each combination of partitioning scheme and tree prior, using stepping-stone sampling with 25 path samples (*Xie et al., 2011*).

The evolutionary timescale was estimated using a strict clock model, with the sampling times of the mitogenomes serving as calibrations for the clock (*Rambaut et al., 2016*). A uniform prior of $(10^{-10}, 10^{-4})$ was used for the mutation rate, with a separate rate assigned to each subset of the data defined by the partitioning scheme. This approach is consistent with previous analyses of time-structured mitogenomic data sets (e.g. *Anijalg et al., 2018*).

Posterior distributions of parameters were estimated by Markov chain Monte Carlo (MCMC) sampling. Samples were drawn every 5000 steps from a chain with a total length of 50,000,000 steps. The MCMC analysis was run in duplicate to check for convergence and the first 10% of samples were discarded as burn-in. Effective sample sizes of the model parameters were estimated to ensure that they were all over 200, which indicates sufficient sampling.

## Appendix 7

### TempNet age categories

Age categories were chosen based on changes in climate and hunting pressure. Samples were divided into four groups (*Supplementary file 1e*): >12,000 years old (i.e., Late Pleistocene samples); 1,000–12,000 years old (i.e., Holocene samples when hunting pressure was low and opportunistic); ~500 years old (i.e., the period in which intense hunting began but when diversity should be representative of the previous 12,000 years); and <250 years old (i.e., samples from during the period of intense hunting, including samples from the last reliably seen pair, killed in 1844). For samples with available date information, the median age was used to determine age group. The 16 samples without date information were placed in the most appropriate group based on other information that allowed us to estimate their ages. For example, the samples from Funk Island are unlikely to be over 1000 years old and are most likely to be around 500 years old.

## Appendix 8

# Population viability analysis: Details and justification (See also *Supplementary file 2a, 2b and 2c*)

Simulation scenarios were set to run for a 350 year period, as intense hunting began in ~1500 AD (*Bengtson, 1984*; *Fuller, 1999*; *Gaskell, 2000*; *Steenstrup, 1855*) and no confirmed sightings of great auks occurred later than 1852 (*BirdLife International, 2016a*; *Fuller, 1999*; *Grieve, 1885*). Data produced in this study show a lack of population structuring in the great auk (see *Figure 3*), and we therefore consider great auks of the North Atlantic to form a single panmictic population. Scenarios were run as a population-based model. Models were also run under scenarios with various definitions of extinction to evaluate any impacts on our results. Extinction was defined as: only one sex remains; population size below the critical limit of 50; or population size below the critical limit of 500. These values are based on the '50:500 rule', which refers to a species' risk of extinction as defined by *Franklin (1980)*.

The outcomes of our simulations were unaffected by the choice of definition used for extinction, which is unsurprising because hunting pressure did not cease towards the extinction of the species. Given this hunting pressure, even 500 birds were well below the sustainable population size, so independent of whether the population size declined to 500 or 50 birds or there was just one sex remaining, the species was bound for extinction. These results might have looked different if our simulations had assumed a complete cessation of hunting when only 500 or only 50 birds were remaining. However, the historical record clearly shows that this was not the case. In fact, as the rarity of the great auk increased, it became more desirable for inclusion in private and institutional collections, as was the case for the last breeding pair killed on Eldey Island in June 1844 (*Bengtson, 1984*; *Fuller, 1999*; *Gaskell, 2000*; *Grieve, 1885*; *Newton, 1861*; *Steenstrup, 1855*; *Thomas et al., 2017*). All results reported are from simulations run under extinction defined as 'only one sex remains'.

Age of first breeding for the great auk is estimated to be 4–7 years old (*Bengtson, 1984*), and a conservative value of 4 years was therefore adopted for the model. The younger the age of first breeding, the less susceptible to extinction the species is. The species is assumed to have been monogamous, laying only one egg per breeding season,, and it is thought they did not replace the egg if it was lost (*Bengtson, 1984*; *Birkhead, 1993*; *Fuller, 1999*). An assumed sex ratio of 1:1 has been applied. Life expectancy is estimated to have been 20–25 years (*Bengtson, 1984*) and we assume that breeding remained possible until death. As several alcid species breed annually once they reach sexual maturity (*De Santo and Nelson, 1995*), we set reproductive rate to 100% adult females breeding and all females producing exactly one egg per year.

Mortality rates were estimated based on records from extant alcids. *De Santo and Nelson (1995)* report survival rates for alcid species at various life stages. Mortality at age 0–1 includes hatchlings and fledglings. For the great auk, we estimate mortality to be 9% (SD: 1), consistent with the lowest mortality reported for any alcid species for this age category by *De Santo and Nelson (1995)* (Japanese murrelet, *Synthliboramphus wumizusume*). With regard to the simulation model, juvenile mortality includes mortality in the age groups 1–2, 2–3, and 3–4; therefore, our juvenile mortality rate was divided between these groups. The lowest mortality for this age group reported by *De Santo and Nelson (1995)* is that of the crested auklet (*Aethia cristella*; 34%). This corresponds to approximately 13% (SD: 1) mortality per year over three years, if the population size of the respective previous year is used as reference in each year. Annual adult survival rate is estimated to be quite high for great auks, because of their large size (*Bengtson, 1984*; *Montevecchi and Kirk, 1996*). Annual adult survival in other alcids is also high, with the razorbill being the highest reported at 93% (*De Santo and Nelson, 1995*). We therefore used an annual mortality rate of 7% (SD:1) for adult great auks. Strictly applying the rule that we use the lowest mortality rate of any alcid species found in the literature leads to some settings that are questionable from a biological perspective. For example, our 0–1 year hatchling mortality is lower than our 1–4 years juvenile mortality. However, as we have no information about actual mortality rates in great auks, any

adjustment of these settings would be arbitrary. We therefore chose to strictly use the lowest mortality rates found in the literature for each age class. For comparison, we added a simulation based on known mortality rates of the razorbill, which have a more biological realistic distribution of mortality rates, albeit perhaps somewhat too high for the great auk (see Discussion and *Supplementary file 2a*).

A reduction of population size, even by harvesting, might have a positive effect on reproductive rate and mortality by freeing up resources and reducing competition. As our reproductive rate was already 100%, a way to simulate such effects was to introduce a linear, density-dependent reduction of mortality rates to half the initial value, following the formula: (0.5+(0.5*PS1))*[initial mortality rate], with PS1 being defined as initial population size (N) divided by carrying capacity (K). Simulations were run with and without this density-dependent reduction in mortality rates (DD) (*Supplementary file 2b*).

We initially estimated the census size (Nc) for our population viability analyses from our estimated effective female population size (Ne) by doubling the effective female population size and dividing the result by Ne/Nc rations typical for birds as summarized by *Frankham (1995)*. However, the range of known, typical Ne/Nc ratios for birds extends over two orders of magnitude, from 0.052 to 0.74 (*Frankham, 1995*). Given these ratios, our estimates for the census size of great auks ranged from 12,292 to 756,346. As we did identify a Pleistocene population bottleneck, and given the large population size reported in historic sources, the actual census size was likely close to or even higher than the upper margin of these estimates, and this is consistent with census sizes currently estimated for the great auk's closest relative, the razorbill (*Alca torda*). Within a range similar to that of the great auk, the IUCN Red List estimates that the razorbill (*Alca torda*) currently has a population size of 979,000–1,020,000 mature individuals. Within the same range, the common murre (*Uria aalge*) and the thick-billed murre *Uria lomvia*) are estimated to have population sizes of 2,350,000–3,060,000 and 1,920,000–2,840,000 mature individuals, respectively (*BirdLife International, 2016b*; *BirdLife International, 2016c*; *BirdLife International, 2017*). Therefore, we conservatively aimed for mature population sizes of 1,000,000 and 3,000,000 great auks.

Razorbills and murres can fly and therefore have access to a larger number of breeding sites than the great auk. They are also much smaller birds, which could facilitate larger population sizes in the same range. On the other hand, razorbill and murre populations may be more affected by hunting today than great auk populations were at the time intensive hunting started. Overall, we feel that our population-size estimates are a reasonably realistic reflection of great auk population sizes. We used Vortex 10.2.8.0 to estimate the census size from the number of mature individuals, assuming that birds reach maturity at 4 years of age, that they show a stable age distribution, and that the different age classes follow the fixed mortality rates described below. This resulted in census sizes for our simulations of 2,000,000 and 6,000,000 birds respectively.

To estimate hunting pressure, we compared models in which various proportions of the population were harvested (see *Supplementary file 2c* for example of how harvest rates were calculated). The age categories for harvest rate are 0–1, 1–2, 2–3, 3–4, and over four for both males and females. We allocated 75% of the harvest rate to the over four category as it was assumed that predominantly adult birds were harvested due to being easily accessible when breeding. The remaining 25% was then split evenly between the other four age categories (0–1, 1–2, 2–3, 3–4), as although it has been reported that young were used as bait (*Grieve, 1885*), it is unlikely they were harvested at the same intensity as the adults and represented a smaller proportion of the overall population.

As we know eggs were collected as well, we allowed for this in the model. The harvest rate for eggs was set at 5%, corresponding to 25,688 and 77,065 respectively for the two initial population sizes tested in our simulations. As great auks nested in dense groups (*Bengtson, 1984*) eggs would have been easy to collect. Based on estimates of breeding pairs at Funk Island (>100,000) (*Birkhead, 1993*), these two values allowed us to test the impact of a quarter or three quarters of all the eggs laid annually on Funk Island alone being harvested, with no harvest occurring anywhere else in the great auk range. We also ran simulations with no egg harvesting to evaluate whether this significantly changed our conclusions. With these

egg harvest settings, an annual bird harvest rate of 10% of the number of birds in the pre-hunting population was identified as critical limit, with significant numbers of simulations leading to extinction. At 10.5% bird harvest rate, all simulations that included egg harvesting and a significant proportion of simulations excluding egg harvesting resulted in extinction.

Our comparative simulations with more 'realistic' rather than conservative settings, including razorbill mortality rates were conducted under the settings outlined in *Supplementary file 2a*.

## Appendix 9

## Nuclear SNP data

As the results of our mitochondrial genome revealed a lack of population genetic structure and high genetic diversity in the great auk, we attempted to target nuclear DNA (nuDNA) to further investigate these results, and to obtain a more detailed picture of great auk evolution and extinction. Initially, twelve samples were chosen for capture of 495 nuclear markers.

Samples were chosen based on the percentage of reads retained in preliminary mitogenome capture dataset, as a rough indication for sample preservation and quality, as well as their geographical location to represent individuals from as much of the former distribution as possible. DNA extraction and library preparation methods were as described for the mitogenome work (see Materials and methods main text). Great auk shotgun genome data (Gilbert et al., bait design available see *Source data 1*) mapped against the razorbill genome (Feng et al. In Review) were used as data basis for bait design. Target gene regions for hybridisation capture enrichment were selected using the following filters:

Paralog genes were excluded from the capture by using UniProtIDs and EnsemblIDs in the razorbill (*Alca torda)* annotation (Feng et al. In Review).

Genes that were missing coverage for more than 20% of their length when mapping great auk reads against the razorbill genome were excluded.

Great auk consensus genes were generated by replacing the razorbill genes with the homozygous SNPs found in great auk.

Genes with the highest percentage divergence between the razorbill and great auk, that didn't contain any N's in their sequence, and which were less than 5kbps in length, were used to build the 20K probes resulting in 495 genes.

MYcroarray probes of 120 bps long with 3x tiling (40 bps shifts) were made from CDS regions and intron regions that were adjacent to the exons of the 495 genes. Enrichment for nuclear genes was performed using MYcroarray MYbaits, following the MYcroarray Mybaits manual v3 (MYcroarray/*MYcroarray MYbaits, 2016*), using 24 hr hybridisation time, at 65°C and final elution into 30 µl nuclease free water. Samples were sequenced on an Illumina MiSeqPE75 platform by New Zealand Genomics Limited, Otago.

Sequencing reads were processed using the PALEOMIX v1.2.5 pipeline (*Schubert et al., 2014*) following a procedure similar to that described by the authors. Briefly, we used AdapterRemoval v2.1.17 (*Schubert et al., 2016*) to trim the reads for adapters and low quality bases (BaseQ <5 or Ns), and to exclude those reads shorter than 30 bp or with more than 50 bp of missing data. Filtered reads from each sample were mapped against the razorbill reference genome (Gilbert, unpublished) using BWA-MEM v0.7.12 (*Li, 2013*), and those with low mapping quality (MapQ <15) removed. After the initial alignment step, Picard (v1.128, https://broadinstitute.github.io/picard) was used to exclude reads that were PCR or optical duplicates. Subsequently, GATK v3.5.0 (*McKenna et al., 2010*) was used to perform a realignment step around indels. As we are dealing with historical samples, we also quantified the extent of DNA damage in our samples using mapDamage v2.0.6 (*Jónsson et al., 2013*). We characterised rates of deamination in double strands (DeltaD) and single strands (DeltaS), as well as the probability of reads not terminating in overhangs (Lambda, transformed into 1/Lambda − 1, a proxy for the overhang length of overhanging regions). From these analyses, we also rescaled base quality scores according to the probability of each base being affected by post-mortem damage.

Read processing of the twelve samples initially sequenced revealed low coverage of both the 495 targeted markers (0.0018x MK78 - 1.2592x MK134), and the razorbill genome overall (0.00006x MK83 - 0.0190x MK50) (*Appendix 9—table 1* and *Appendix 9—table 2*). Only one sample, MK134, had any genes with at least 3-fold coverage (*Appendix 9—figure 1*). Therefore, further analysis that would provide any meaningful results could not be performed.

**Appendix 9—table 1.** Estimated coverage information from the twelve sequenced samples. The estimated coverage of the 495 targeted genes and estimated coverage of the reads that mapped to the razorbill genome is reported.

| Sample | Country | Estimated coverage of razorbill genome | Estimated coverage of targeted genes |
|---|---|---|---|
| MK49 | Norway | 0.0101 | 0.0152 |
| MK50 | Iceland | 0.0190 | 0.0155 |
| MK78 | Funk Island | 0.0022 | 0.0018 |
| MK83 | Funk Island | 0.00006 | 0.0071 |
| MK103 | Funk Island | 0.0011 | 0.0150 |
| MK106 | Sweden | 0.0172 | 0.0105 |
| MK115 | Norway | 0.0012 | 0.0021 |
| MK131 | Iceland | 0.0090 | 0.0423 |
| MK133 | Skin Mystery | 0.0190 | 0.0154 |
| MK134 | Skin Mystery | 0.0179 | 1.2592 |
| MK135 | Skin Mystery | 0.0073 | 0.0106 |
| MK136 | Skin Mystery | 0.0021 | 0.0128 |

**Appendix 9—table 2.** Coverage range of captured markers. Numbers in square brackets represent the number of markers which have 0 coverage. Genes with the highest coverage are shown in brackets.

| Sample | Country | Coverage range of captured markers |
|---|---|---|
| MK49 | Norway | 0 [125] – 0.4898 (Fam174b) |
| MK50 | Iceland | 0 [157] – 0.2204 (Isca2) |
| MK78 | Funk Island | 0 [379] – 0.1087 (Mrp130) |
| MK83 | Funk Island | 0 [223] – 0.2960 (Nipbl) |
| MK103 | Funk Island | 0 [164] – 0.7049 (Glrx5) |
| MK106 | Sweden | 0 [190] – 0.2403 (Pcp4) |
| MK115 | Norway | 0 [366] – 0.2263 (Tmem60) |
| MK131 | Iceland | 0[78] – 1.5238 (Ssna1) |
| MK133 | Skin mystery | 0 [129] – 0.3061 (Fam174b) |
| MK134 | Skin mystery | 0.0628 (TPK1) – 17.7232 (Ssna1) |
| MK135 | Skin mystery | 0 [172] – 0.2580 (myct1) |
| MK136 | Skin mystery | 0 [142] – 0.4067 (myct1) |

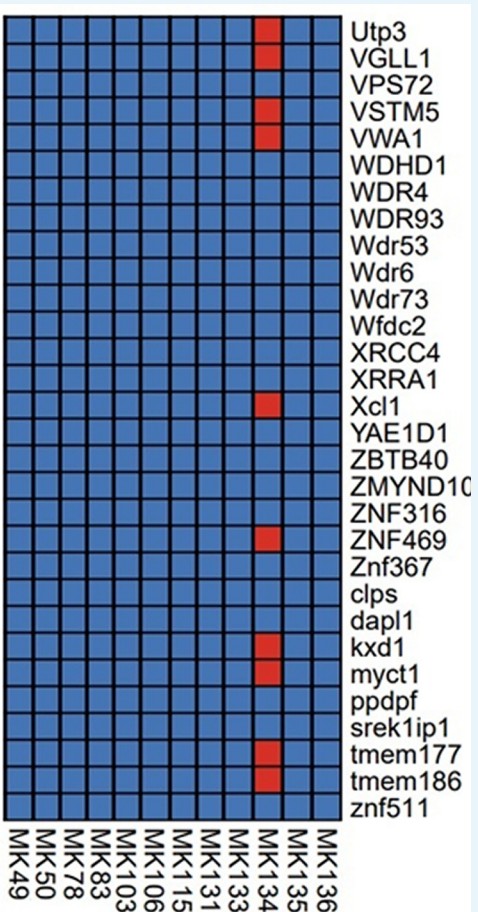

**Appendix 9—figure 1.** Section of the presence/absence matrix showing coverage of 30/495 captured genes (listed on the right-hand side) for each sample sent for sequencing. Presence is defined as coverage >= 3, indicated by a red square, absence is indicated by a blue square.

**Appendix 10**

## Additional analyses Spanish samples

In the phylogenetic tree of Great Auks that yielded sufficient sequence data as per our filtering criteria (Appendix 5) the sample MK40_Spain appeared to be differentiated from the rest of the samples which came from the northern regions of their distribution. This raises the question whether the sample could represent a refugial population in Spain. In order to test this, we re-examined the phylogenetic relationships between samples with the addition of the other Spanish samples we sequenced but which did not fulfil the filtering criteria for inclusion in our final dataset. These samples included MK37, MK42, MK44 and MK45. These samples were characterised by poor coverage (average coverage ranged from 0.18 to 2.07) and over 33% of bases missing from consensus sequence (consensus sequence length ranged from 36 bp to 5468 bp). Sequences generated as described using the Paleomix pipeline (Appendix 5) for samples MK37, MK42, MK44 and MK45 were manually aligned to the reference genome using Bioedit v7.2.5 (*Hall, 1999*) and Tablet v1.16.09.06 (*Milne et al., 2013*) to view the rescaled Binary Alignment Map (BAM). As MK37 and MK45 were of very poor quality, we were unable to use them in this additional analysis. However, we were able to produce an alignment of 859 bp that included the additional Spanish samples MK42 and MK44. A Neighbour-joining analysis of the alignment based on p-distances (*Saitou and Nei, 1987*) using MEGAX (*Kumar et al., 2018*) yielded a very poorly resolved phylogeny (*Appendix 10—figure 1*). Critically, the new Spanish samples do not group with the outlier MK40, thereby not supporting a hypothesis of a Spanish refugial population.

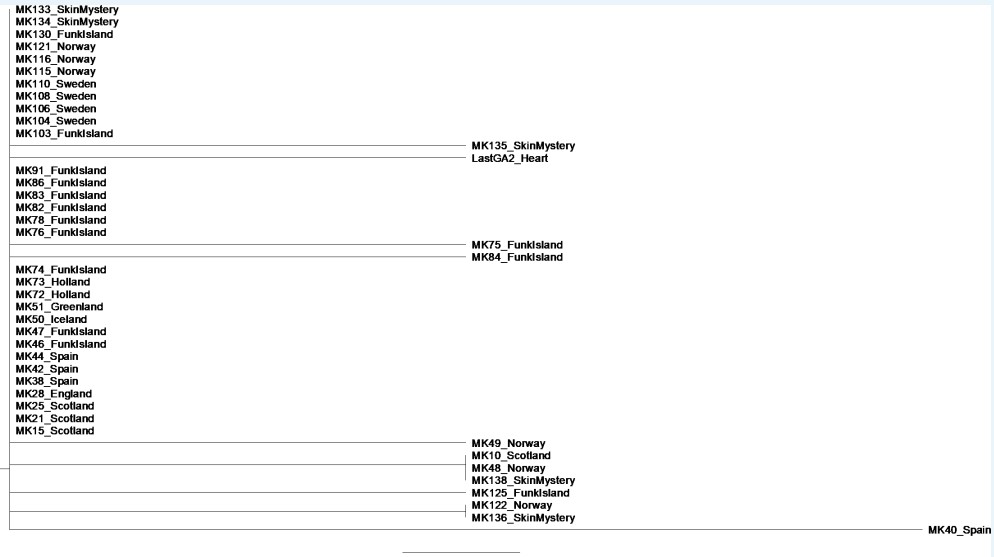

**Appendix 10—figure 1.** Phylogenetic tree showing the relationship between all samples that passed filtering criteria, plus additional Spanish samples previously excluded from analysis. The evolutionary history was inferred using the Neighbor-Joining method (*Saitou and Nei, 1987*) in MEGAX (*Kumar et al., 2018*). The optimal tree with the sum of branch length = 0.01164144 is shown. The percentage of replicate trees in which the associated taxa clustered together in the bootstrap test (1000 replicates) are shown next to the branches (*Felsenstein, 1985*). The tree is drawn to scale, with branch lengths in the same units as those of the evolutionary distances used to infer the phylogenetic tree. The evolutionary distances were computed using the p-distance method (*Nei and Kumar, 2000*) and are in the units of the number of base differences per site. This analysis involved 43 nucleotide sequences. All positions containing gaps and missing data were eliminated (complete deletion option). There were a

total of 859 positions in the final dataset. Tip labels give the sample names and sampling locations.

