## [Decision Letter]

**Acceptance summary:**

This paper reports an exciting attempt to draw new inferences about the extinction history of the Great Auk. Generating a large number of mitogenomes from an extinct species requires considerable effort, and has so far only been achieved for very few taxa (e.g., mammoth and cave bear). The reviewers noted that additional nuclear markers would have significantly strengthened analyses, but appreciated that obtaining this information proved methodologically difficult. While some uncertainty remains, this study represents an important step in advancing our understanding of humans' role in the extinction of the iconic Great Auk.

**Decision letter after peer review:**

Thank you for submitting your article "Demographic reconstruction from ancient DNA supports rapid extinction of the Great Auk" for consideration by *eLife*. Your article has been reviewed by four peer reviewers, and the evaluation has been overseen by Christian Rutz as Reviewing Editor and Ian Baldwin as Senior Editor. The following individual involved in the review of your submission has agreed to reveal their identity: Paul Wade.

The Reviewing Editor has consolidated the reviewers' feedback and drafted this decision letter, to help you prepare a revised submission.

Essential revisions:

1) Study assumptions:

The reviewers noted that some aspects of the study inevitably remained quite speculative, and felt that this should be better acknowledged throughout. Therefore, please state explicitly the assumptions, and degree of certainty, for all results and conclusions; all key information should be contained in the main text, and not just the online supplement.

2) GPS Drift capsules:

There was broad agreement that this was one of the weaker aspects of the study. Specifically, it is not clear whether currents can be reliably mapped with just two capsules, and whether it is possible to use these data to make robust inferences about currents thousands of years ago. Furthermore, are currents assumed to be stable throughout the year? Adult Great Auks would presumably have only moved at certain times of the year, and because they were almost certainly philopatric, the map's uni-directional arrows raise the question of how they returned to their colonies. Please tone down the inferences drawn from the drift capsule dataset, highlight ocean current modelling as a challenge for future work, and provide better justification for this line of enquiry (the Great Auk was a flightless bird and therefore may have relied on, or at least been affected by, ocean currents).

3) Genomic analyses:

The reviewers have raised several issues about the genomic analyses that need addressing.

a) There is a hint that a Spanish sample may have been differentiated from the Northern ones, and could represent a refugial population. Were there any partial mtDNA data from the other 8 samples from Spain that could be used to explore this further?

b) It is unfortunate that the Funk Island samples had no date and were excluded from analyses, although their addition with a wide prior had little effect on results; please add the corresponding tree to the online supplement (like Appendix—figure 4). Are there no additional historical museum samples, or feather specimens, available that might yield more and better quality sequence data than the old subfossil bones?

c) The complete lack of phylogeographic structure is surprising. Such lack of structure is usually explained either by high dispersal or a relatively recent bottleneck, followed by range expansion. Since the Great Auk was a flightless bird, it would be very interesting if you could add a comparison to the phylogeographic structure (or lack thereof) found in penguins (e.g., Discussion, third paragraph).

d) Related to the previous comment, the manuscript could be improved by a brief comparison, perhaps in the Discussion, with what previous studies have found when analysing temporally sampled mtDNA data across demographic declines (e.g., in either extinct species or species that are critically endangered today). The manuscript does not appear to cite any papers that analysed genetic changes across the time scales explored here (i.e., a few centuries). Perhaps the human-caused decline in the Great Auk was even more rapid than the declines that have been observed in many species during the last century? Do you have enough samples (i.e., power) to detect a decline in genetic diversity in the last 100 years prior to the Great Auk's extinction?

e) Supplementary Figure 4: Please add a higher-resolution version of the TempNet figure as a main figure, instead of as a supplementary item. The reason for this is partly that there is currently no easily accessible information in the main text on how the samples are temporally distributed, which is important for context. Furthermore, the finding that diversity was still high in the last time bin (<250 years) is very interesting, and should be discussed in more detail.

4) Population viability analyses:

The reviewers generally found the population viability analyses (PVA) well implemented, but highlighted a few points that need addressing.

a) You chose to specify a fairly optimistic life history scenario for the Great Auk, reasoning that, if the model predicts extinction even under these conditions, it can be safely concluded that extinction risk must have been high in the real world. While this is a reasonable approach, the PVA settings were perhaps a touch too optimistic, with zero environmental variation in reproduction, and very low variation in mortality rates (set at 1% for all age classes). Furthermore, it seems that Vortex was run with harvest rate set as a constant percentage of population size (please clarify in the revision whether this was actually the case, as it was hard to tell from either the manuscript or the Vortex documentation), even though it is conceivable that humans may have continued harvesting a constant number (at least for a while), as it would have been relatively easy to find the birds on their nesting colonies. When run on such basic settings, the PVA comes close to being a deterministic model, and a simple lambda calculation could have yielded the same insight: using life history parameters from Supplementary file 2A and B, lambda can be estimated as ~1.12, and one would conclude that a harvest rate of 12% would clearly be unsustainable in a deterministic manner. Please consider exploring a more realistic suite of settings, and provide some further context to justify your PVA approach.

b) In general, the life history parameters seem carefully chosen, but it is slightly odd that annual juvenile mortality (13%) is higher than hatchling/fledgling mortality (9%). Can this be justified, or adjusted? Furthermore, estimates for age of first breeding and adult survival probably need amending.

c) Is there more historical information on harvest rates, such as numbers of birds or eggs taken from a particular nesting colony, that would allow the calculation of site- and time-specific harvest rates?

---

## [Author Response]

Essential revisions:1) Study assumptions:The reviewers noted that some aspects of the study inevitably remained quite speculative, and felt that this should be better acknowledged throughout. Therefore, please state explicitly the assumptions, and degree of certainty, for all results and conclusions; all key information should be contained in the main text, and not just the online supplement.

Thank you for this feedback. Where appropriate, we have tried to improve the clarity of our assumptions, the degree of certainty of various results and conclusions. Specifically, we discussed in more detail our conclusions of a rapid population decline and what our data does and does not show (first paragraph of the Discussion). We have also more explicitly identified speculations in the discussion of the GPS data. Furthermore, we have added vortex simulations with “realistic” mortality and reproduction rates derived from the razorbill to put our conservative sustainable harvest rate estimates into context and highlight the uncertainty of our estimates. Finally, we moved the TempNet figure into the main text as requested by the reviewers.

2) GPS Drift capsules:There was broad agreement that this was one of the weaker aspects of the study. Specifically, it is not clear whether currents can be reliably mapped with just two capsules, and whether it is possible to use these data to make robust inferences about currents thousands of years ago. Furthermore, are currents assumed to be stable throughout the year? Adult Great Auks would presumably have only moved at certain times of the year, and because they were almost certainly philopatric, the map's uni-directional arrows raise the question of how they returned to their colonies. Please tone down the inferences drawn from the drift capsule dataset, highlight ocean current modelling as a challenge for future work, and provide better justification for this line of enquiry (the Great Auk was a flightless bird and therefore may have relied on, or at least been affected by, ocean currents).

We have amended the manuscript in several sections that discuss the GPS capsule data. The changes made are as follows:

Results: Ocean current data: To evaluate potential reasons for the observed lack of population structure, we sourced data from GPS-equipped drifting capsules that had been deployed in the North Atlantic as part of the “Message in a Bottle” project by Verkís Consulting Engineers. As the great auk was flightless, ocean currents might have influenced its migration patterns. The route taken by the capsules connects some of the main breeding colonies in St Kilda (Scotland), Geirfuglasker/Eldey Island (Iceland), and Funk Island (Canada) (Figure 4). The extrapolation of present-day ocean current data into the past and the interpretation of the data in the context of great auk movements is merely speculative. However, if ocean currents today are comparable to those of past millennia, then the data do at least provide a possible explanation for how great auks travelled across their former range and between breeding colonies (Figure 4). A full description of the routes taken by the capsules is provided in Appendix 1.

Discussion: We can only speculate what factors may have driven this lack of population structure, but the data collected from the GPS-enabled drifting capsules are consistent with hypotheses put forward by a number of authors. It has been suggested that migrations occurred in both northward and southward directions between breeding and wintering sites, aided by ocean currents such as the East Greenland Current (Brown, 1985; Meldgaard, 1988; Montevecchi and Kirk, 1996). However, as these preliminary data were only available from two GPS-enabled drifting capsules and as ocean currents may have changed significantly over the past few centuries, the conclusions that we can draw from such data are somewhat limited. Furthermore, it is possible that these currents can change throughout the year. Thus, these data must be considered with caution and pending far more detailed studies of ocean currents in the North Atlantic throughout the year. Nevertheless, high vagility of the great auk is further supported by its ability to track its habitat in response to climate change, as evidenced by archaeological records (Bengtson, 1984; Campmas et al., 2010; Meldgaard, 1988; Serjeantson, 2001).

We hope that this now makes it clear that although the data has been included in the Discussion and used in supporting some of our conclusions, it is from a small, preliminary dataset and should therefore be viewed that way.

3) Genomic analyses:The reviewers have raised several issues about the genomic analyses that need addressing.a) There is a hint that a Spanish sample may have been differentiated from the Northern ones, and could represent a refugial population. Were there any partial mtDNA data from the other 8 samples from Spain that could be used to explore this further?

Thank you for this observation. We have explored this further as good as possible with limited data (see below). Our results suggest that there is no evidence for a refugial population in Spain. Samples from Spain are similarly spread across the tree as samples from other regions. Below, we have summarized the additional analyses we have conducted. However, as they do not show any pattern that is unobserved in the data we use for this publication, we would prefer to leave this section out of the manuscript. We feel that it would impact the flow and clarity of the manuscript without adding any crucial information. We are of course open to including this section, for example in the appendix, if the editor sees this fit.

Additional analyses Spanish samples.

In the phylogenetic tree of Great Auks that yielded sufficient sequence data as per our filtering criteria (Appendix 4) the sample MK40_Spain appeared to be differentiated from the rest of the samples which came from the northern regions of their distribution. This raises the question whether the sample could represent a refugial population in Spain. In order to test this, we re-examined the phylogenetic relationships between samples with the addition of the other Spanish samples we sequenced but which did not fulfil the filtering criteria for inclusion in our final dataset. These samples included MK37, MK42, MK44 and MK45. These samples were characterized by poor coverage (average coverage ranged from 0.18-2.07) and over 33% of bases missing from consensus sequence (consensus sequence length ranged from 36bp to 5468bp). Sequences generated as described using the Paleomix pipeline (Appendix 4) for samples MK37, MK42, MK44 and MK45 were manually aligned to the reference genome using Bioedit v7.2.5 (Hall, 1999) and Tablet v1.16.09.06 (Milne et al., 2013) to view the rescaled Binary Alignment Map (BAM). As MK37 and MK45 were of very poor quality, we were unable to use them in this additional analysis. However, we were able to produce an alignment of 859bp that included the additional Spanish samples MK42 and MK44. A Neighbour-joining analysis of the alignment based on p-distances (Saitou and Nei, 1987) using MEGAX (Kumar et al., 2018) yielded a very poorly resolved phylogeny (Author response image 1). Critically, the new Spanish samples do not group with the outlier MK40, thereby not supporting a hypothesis of a Spanish refugial population.

**Author response image 1. respfig1:** The evolutionary history was inferred using the Neighbor-Joining method [Saitou and Nei, 1987]. The optimal tree with the sum of branch length = 0.01164144 is shown. The percentage of replicate trees in which the associated taxa clustered together in the bootstrap test (1000 replicates) are shown next to the branches [Felsenstein, 1985]. The tree is drawn to scale, with branch lengths in the same units as those of the evolutionary distances used to infer the phylogenetic tree. The evolutionary distances were computed using the p-distance method [Nei and Kumar, 2000] and are in the units of the number of base differences per site. This analysis involved 43 nucleotide sequences. All positions containing gaps and missing data were eliminated (complete deletion option). There were a total of 859 positions in the final dataset. Evolutionary analyses were conducted in MEGA X [Kumar et al., 2018].

b) It is unfortunate that the Funk Island samples had no date and were excluded from analyses, although their addition with a wide prior had little effect on results; please add the corresponding tree to the online supplement (like Appendix—Figure 4).

We agree that it is unfortunate that the Funk Island samples do not have date information, however, sadly that is the nature of the sample from this site which is essentially a mass grave yard of great auks, with hundreds of bones from the period of intense hunting. It would certainly be interesting future work to be able to carbon date some of the samples from Funk Island to add this information to the data set. However, for now we have added all undated samples to the tree and added it to the supplement as requested (New Appendix—figure 5).

Are there no additional historical museum samples, or feather specimens, available that might yield more and better quality sequence data than the old subfossil bones?

This study was part of a PhD project that was completed in 2017, therefore, unfortunately, funding and personnel to facilitate further sampling is no longer available. During the project extensive sampling was carried out, sourcing samples that represented individuals from as much of the former great auk range as possible, and over as great a period as possible. Samples were collected from number of museums, worldwide. As we wanted to include samples in the dataset which had as much information as possible, we chose not to include many mounted museum specimens as the vast majority of these do not have sample date or sample location information.

c) The complete lack of phylogeographic structure is surprising. Such lack of structure is usually explained either by high dispersal or a relatively recent bottleneck, followed by range expansion. Since the Great Auk was a flightless bird, it would be very interesting if you could add a comparison to the phylogeographic structure (or lack thereof) found in penguins (e.g., Discussion, third paragraph).

Thank you for the suggestion to compare our result to the penguins. It helps to put our findings from the great auk in context and shows that while the lack of population structure in great auks is surprising, it does not actually appear to be unusual in flightless seabirds. We have now added a paragraph that reads: While all of the great auk’s closest relatives are capable of flight, which would aid population connectivity, a lack of population structure has similarly been report from some penguin species. For example, little or no population structure has been reported for the emperor penguin (*Aptenodytes forsteri*) (Cristofari et al., 2016), chinstrap penguin (*Pygoscelis antarcticus)* (Mura-Jornet et al., 2018), and Adélie penguin (*P. adeliae*) Gorman et al., 2017; Roeder et al., 2001).

d) Related to the previous comment, the manuscript could be improved by a brief comparison, perhaps in the Discussion, with what previous studies have found when analysing temporally sampled mtDNA data across demographic declines (e.g., in either extinct species or species that are critically endangered today). The manuscript does not appear to cite any papers that analysed genetic changes across the time scales explored here (i.e., a few centuries). Perhaps the human-caused decline in the Great Auk was even more rapid than the declines that have been observed in many species during the last century? Do you have enough samples (i.e., power) to detect a decline in genetic diversity in the last 100 years prior to the Great Auk's extinction?

This was indeed an obvious omission in our Discussion. We have now added a section to the first paragraph of the Discussion which discussed this point as follows:

If the great auk had been at risk of extinction prior to the onset of intensive human hunting, for example as a result of long-term suboptimal habitat or environmental change, we would expect to see genetic evidence of such stress, as for example observed in studies of cave bears (Stiller et al., 2010) and bison (Shapiro et al., 2004). If, on the other hand, the population declined rapidly, for example as a result of extensive hunting, genetic data would have only very limited power to detect such a decline in a long-lived species. Mitochondrial DNA studies of New Zealand moa found no evidence of a population declines prior to extinction (Allentoft et al., 2014; Rawlence et al., 2012) and a study of the endemic Hawaiian Petrel came to a similar conclusion (Welch et al., 2012). In fact, even a recent whole-genome study of two extinct New Zealand songbirds (huia and South Island kõkako), which disappeared after human settlement within 700 years, found no genetic evidence of population decline prior to the disappearance of the species (Dussex et al., 2019). Therefore, our results are consistent with a rapid decline of great auks. It is important to keep in mind, though, that our results simply indicate that the demise of the great auk was beyond the detection limit of genetic data. They do not necessarily confirm whether the rapid demise that must have taken place prior to extinction started before or after the onset of extensive human hunting, nor do the results provide an indication of whether there was more than one population decline. A localised, unexplained decline in great auk numbers on the eastern side of the North Atlantic over the past 2,000 years, for example, which has been inferred from a decline in bone finds in England, Scotland, and Scandinavia (Bengtson, 1984; Best and Mulville, 2014; Grieve, 1885; Hufthammer, 1982; Serjeantson, 2001), does not appear to have been severe enough to leave a genetic signature.

e) Supplementary Figure S4: Please add a higher-resolution version of the TempNet figure as a main figure, instead of as a supplementary item. The reason for this is partly that there is currently no easily accessible information in the main text on how the samples are temporally distributed, which is important for context. Furthermore, the finding that diversity was still high in the last time bin (<250 years) is very interesting, and should be discussed in more detail.

We have added a higher-resolution image of the TempNet figure and moved this to the main text (Figure 2). We hope that the resolution of the image now meets your requirements.

4) Population viability analyses:The reviewers generally found the population viability analyses (PVA) well implemented, but highlighted a few points that need addressing.a) You chose to specify a fairly optimistic life history scenario for the Great Auk, reasoning that, if the model predicts extinction even under these conditions, it can be safely concluded that extinction risk must have been high in the real world. While this is a reasonable approach, the PVA settings were perhaps a touch too optimistic, with zero environmental variation in reproduction, and very low variation in mortality rates (set at 1% for all age classes).

To provide some comparative data with more “realistic” settings, we have now added a simulation with mortality rates derived from the razorbill (including higher standard variation), an increased reproductive age (from 4 to 5 years) and a reduced maximum reproductive age (down to 20 years from 25) See Supplementary file 2B). These simulations yield a sustainable harvest rate of approximately 40,000 birds a year without any egg harvest. This is likely an underestimation, as the razorbill does have a higher reproductive rate than the great auk, and can therefore tolerate a higher mortality. This simulation and its results are now introduced and discussed in appropriate sections throughout the manuscript.

Furthermore, it seems that Vortex was run with harvest rate set as a constant percentage of population size (please clarify in the revision whether this was actually the case, as it was hard to tell from either the manuscript or the Vortex documentation), even though it is conceivable that humans may have continued harvesting a constant number (at least for a while), as it would have been relatively easy to find the birds on their nesting colonies. When run on such basic settings, the PVA comes close to being a deterministic model, and a simple λ calculation could have yielded the same insight: using life history parameters from Supplementary fie 2A and B, λ can be estimated as ~1.12, and one would conclude that a harvest rate of 12% would clearly be unsustainable in a deterministic manner.

Sorry, this was not well described in our manuscript. We actually did use constant numbers rather than a constant percentage. We have now clarified this where appropriate throughout the manuscript.

Please consider exploring a more realistic suite of settings, and provide some further context to justify your PVA approach.b) In general, the life history parameters seem carefully chosen, but it is slightly odd that annual juvenile mortality (13%) is higher than hatchling/fledgling mortality (9%). Can this be justified, or adjusted?

The reasoning behind the somewhat odd distribution of mortality rates was that we wanted a rigorous way to decide on rates in the absence of biological information. Hence we decided on the criterium of “lowest rates of any alcid found in the literature”. And the lowest rates we could find for the 0-1 year timeframe are lower than those for the later time frames. We could have increased the 0-1 values, but it would have been arbitrary, so we decided against it. However, we have now also included analyses with the complete razorbill set of mortality rates and higher standard variations to provide a comparison how our simulations would look with less conservative, but still species derived settings. We have also added a further explanation of our reasoning behind the conservative values into the manuscript which reads (Appendix 7):

“Strictly applying the rule that we use the lowest mortality rate of any alcid species found in the literature leads to some settings that are questionable from a biological perspective. For example, our 0-1 year hatchling mortality is lower than our 1-4 years juvenile mortality. However, as we have no information about actual mortality rates in great auks, any adjustment of these settings would be arbitrary. We therefore chose to strictly use the lowest mortality rates found in the literature for each age class. For comparison, we added a simulation based on known mortality rates of the razorbill, which have a more biological realistic distribution of mortality rates, albeit perhaps somewhat too high for the great auk (see Discussion and Supplementary file 2B).”

Furthermore, estimates for age of first breeding and adult survival probably need amending.

We have now done that in our “realistic settings” analyses (see Supplementary file 2A and B).

c) Is there more historical information on harvest rates, such as numbers of birds or eggs taken from a particular nesting colony, that would allow the calculation of site- and time-specific harvest rates?

Unfortunately, we could not find such information despite extensive literature searches. This was in fact the main reason why we decided on running the analyses with unrealistically conservative settings.